# Reasoning Compartmentalization:
# Bridging the Concretization Gap via Abstraction-based Routing

**Ling-I Wu** [1] [2]  **Jian Tong** [2]  **Yu Sun** [2]  **Xuan Gao** [3]  **Xu Guo** [2] [3]  **Qipeng Guo** [2]  **Kai Chen** [2]  **Guoqiang Li** [1]

## Abstract

While previous research has documented the sensitivity of Large Language Models (LLMs) to surface-level performance degradation, the underlying impact on internal representations and learning dynamics remains under-explored. In this work, we study this question using a controlled setup with paired reasoning tasks that are logically identical but expressed either in an abstract formal language (FL) or in natural language (NL). We find that converting FL problems into NL consistently degrades reasoning accuracy. More importantly, we show that FL and NL inputs activate largely separate internal representations and exhibit weak learning transfer between them. We refer to this phenomenon as **reasoning compartmentalization**. To test whether this compartmentalization can be mitigated, we introduce abstraction-based alignment, where models are trained to translate NL inputs into their corresponding FL forms. While this significantly improves reasoning performance, FL and NL representations remain largely distinct, and learning transfer across formulations remains limited. Through activation-level interventions, we further show that performance improvements arise not from representational fusion, but from improved routing. This suggests that abstraction alleviates formulation sensitivity by strengthening connections between formulation-specific reasoning pathways, rather than by aligning their representations. Code is available on https://github.com/edithwuly/Reasoning-Compartmentalization.

[1]Shanghai Jiao Tong University, Shanghai, China [2]Shanghai Artificial Intelligence Laboratory, Shanghai, China [3]Fudan University, Shanghai, China. Correspondence to: Guoqiang Li <li.g@sjtu.edu.cn>, Qipeng Guo <guoqipeng@pjlab.org.cn>.

*Proceedings of the $43^{rd}$ International Conference on Machine Learning*, Seoul, South Korea. PMLR 306, 2026. Copyright 2026 by the author(s).

## 1. Introduction

The emergence of large language models (LLMs) has underscored their strong performance across a broad range of reasoning tasks. Nevertheless, growing evidence shows that seemingly minor changes in input formulation can substantially undermine this performance, revealing a lack of robustness that complicates reliable deployment in complex, real-world settings. Prior studies identify logically equivalent inputs that differ in surface form, showing that rephrasing (Zhou et al., 2024), typos (Gan et al., 2024), language changes (Hu et al., 2025), and extended contexts (Xu et al., 2025) can all degrade reasoning on established benchmarks. However, most existing studies frame formulation sensitivity as an inference-time artifact: surface variations may perturb outputs, but are not assumed to fundamentally alter the model's internal reasoning process or the way reasoning knowledge is acquired through learning.

In this work, we revisit formulation sensitivity under a strictly controlled setting. We construct paired reasoning tasks expressed in abstract formal language (FL) and concretized natural language (NL) that are logically isomorphic. Each pair shares identical variables, constraints, solution spaces, and correctness criteria, and is near-equivalent for human solvers. Despite this equivalence, we uncover systematic and structured discrepancies across representations along three distinct dimensions. First, reasoning accuracy consistently degrades when moving from FL to NL representations. Second, the internal representations elicited by FL and NL inputs remain strongly separated across layers. Third, the learning dynamics induced by the two representations exhibit pronounced asymmetry.

Taken together, these behavioral, representational, and learning-level discrepancies cannot be attributed to surface noise or differences in task difficulty alone. To explain this pattern, we propose the hypothesis of Reasoning Compartmentalization. Under this hypothesis, logically equivalent tasks expressed in different surface forms induce distinct, representation-conditioned computational channels within LLMs. As a consequence, both inference and optimization become representation-dependent: reasoning knowledge acquired through one channel is not readily accessible from another, leading to fragmented reasoning behavior and poor

cross-representation transfer.

Importantly, we show that while reasoning compartmentalization persists, its impact on learning transfer is not immutable. Through translation-based abstraction training, where models are trained solely to translate NL representations into their corresponding FL representations, without any task-solving supervision, we observe consistent improvements in reasoning performance on both representations. Moreover, following this abstraction training, subsequent single-representation training exhibits substantially strengthened cross-representation transfer.

Strikingly, this restoration in reasoning performance does not arise from representational fusion. Despite substantial gains in accuracy, FL and NL representations remain separated in representation space, and their learning dynamics remain only weakly coupled. To probe the underlying mechanism, we conduct activation-level interventions by selectively exchanging hidden states between FL and NL inputs. These interventions selectively improve prediction specificity, revealing that abstraction training enhances the accessibility of reasoning pathways across representations rather than enforcing representational alignment.

Overall, our findings suggest that, at the representational, learning, and computational levels, reasoning problems that are logically identical but expressed in different formulations can induce fragmented behavior in LLMs, manifested as separated representation spaces, unstable optimization transfer, and impaired computational accessibility across formulations. Abstraction mitigates these failures by restoring routing accessibility between representation-conditioned reasoning channels, rather than collapsing or aligning their internal representations.

In summary, our contributions are fourfold:

- We introduce a controlled abstraction–concretization evaluation framework based on logically isomorphic FL/NL task pairs, enabling direct comparison between symbolic constraint formulations and natural language puzzle formulations.

- We show that concretization induces systematic fragmentation in LLM reasoning: even for logically identical tasks, FL and NL formulations exhibit consistent accuracy gaps and remain strongly separated in representation space across layers.

- We reveal a distinct form of isolation at the level of learning dynamics. Optimization on one representation induces weak, unstable, and non-proportional improvements on the other, indicating that reasoning knowledge acquired under one formulation is not coherently shared during training.

- We show that translation-based abstraction improves reasoning and cross-representation transfer by strengthening routing accessibility across representation-conditioned reasoning channels, while leaving representational separation largely intact.

**Conflict of Interest Disclosure.** The authors are not aware of any financial conflicts of interest related to this work.

## 2. Problem Setup and Data Construction

### 2.1. A Unified Logical Task

Our goal is to study how representational choices affect LLMs' logical reasoning, independent of task semantics. To this end, we require a reasoning task whose underlying logical structure can be precisely controlled while admitting realizations in fundamentally different representational spaces. We adopt Boolean satisfiability (SAT) as a canonical abstraction, as its semantics are defined purely by a set of Boolean constraints and are invariant to the language in which these constraints are expressed.

Concretely, the same SAT instance can be equivalently represented as a formal constraint program (e.g., in Z3) or as a natural language puzzle describing the same relationships in prose. While these representations differ drastically in surface structure, compositional form, and available reasoning cues, they encode an identical logical core.

A SAT instance consists of a Boolean formula over a set of variables, and the task is to determine whether there exists an assignment of truth values that satisfies the formula. Throughout this work, we focus on formulas expressed in conjunctive normal form (CNF). As a concrete example, consider the following formula with $2 \times 2$ variables and 4 clauses:

$$F = (\neg A_1 \lor B_1) \land (A_1 \lor \neg B_2) \land (A_1 \lor B_2) \land (\neg A_2 \lor \neg B_2).$$

One satisfying assignment is

$$A_1 = \text{True}, \quad A_2 = \text{False}, \quad B_1 = \text{True}, \quad B_2 = \text{False}.$$

Each SAT instance will be presented in two alternative but logically equivalent representational forms: a Z3 constraint program and a natural language puzzle. Despite their radically different surface realizations and reasoning affordances, all representations specify the same set of Boolean constraints and admit the same satisfying assignments.

SAT can be viewed as a special case of the *constraint satisfaction problem* (CSP), in which all variables are Boolean and all constraints are logical clauses. We therefore extend our formal-language representation to a richer CSP setting in which variables may be of type `bool` or `int`.

| Model | Method | Bool | | | | | | + Int | | + Abel | |
|---|---|---|---|---|---|---|---|---|---|---|---|
| | | 3 × 3 | | 3 × 5 | | 5 × 5 | | | | | |
| | | FL | NL | FL | NL | FL | NL | FL | NL | FL | NL |
| **Qwen3-30B-A3B** | Orig. | 97.6 | 31.6 | 89.4 | 29.8 | 41.4 | 23.2 | 99.4 | 36.4 | 90.6 | 62.6 |
| | Prom. | - | 65.6 (+34.0) | - | 60.2 (+30.4) | - | 29.2 (+6.0) | - | 66.2 (+29.8) | - | 77.8 (+15.2) |
| | RL | - | 87.8 (+56.2) | - | 84.4 (+54.6) | - | 53.4 (+30.2) | - | 80.2 (+43.8) | - | 83.2 (+21.0) |
| **GPT-oss-20B** | Orig. | 85.8 | 74.2 | 62.0 | 49.2 | 20.2 | 13.0 | 97.8 | 70.2 | 84.4 | 47.8 |
| | Prom. | - | 81.0 (+6.8) | - | 59.4 (+10.2) | - | 17.8 (+4.8) | - | 80.4 (+10.2) | - | 54.2 (+6.4) |
| | RL | - | 86.4 (+12.2) | - | 74.2 (+25.0) | - | 24.2 (+11.2) | - | 83.8 (+13.6) | - | 72.8 (+25.0) |
| **DeepSeek-R1** | Orig. | 99.8 | 73.0 | 99.0 | 61.6 | 92.6 | 71.2 | 100 | 83.8 | 97.8 | 63.6 |
| | Prom. | - | 92.2 (+19.2) | - | 81.0 (+19.4) | - | 76.4 (+5.2) | - | 88.4 (+4.6) | - | 71.8 (+8.2) |
| **Gemini-2.5-Pro** | Orig. | 99.2 | 80.2 | 98.8 | 74.0 | 86.6 | 80.2 | 100 | 82.2 | 99.2 | 66.8 |
| | Prom. | - | 89.4 (+9.2) | - | 78.8 (+4.8) | - | 68.6 (-11.6) | - | 87.4 (+5.2) | - | 76.0 (+9.2) |
| **GPT-o3** | Orig. | 99.4 | 97.0 | 99.8 | 97.8 | 99.4 | 98.8 | 100 | 87.0 | 99.8 | 72.4 |
| | Prom. | - | 98.0 (+1.0) | - | 99.6 (+1.8) | - | 99.0 (+0.2) | - | 90.4 (+3.4) | - | 83.6 (+11.2) |

*Table 1.* The accuracy of LLMs on our generated abstraction–concretization paired dataset before (Orig.), after introducing the intermediate prompt-based step (Prom.), and after abstraction-enhanced reinforcement learning (RL).

For example, consider a CSP with

$$x \in \{\text{True}, \text{False}\}, \quad y \in \{0, 1, 2\}, \quad z \in \{0, 1\},$$

and constraints

$$(x = \text{True} \Rightarrow y \leq 1), (y + z = 2), (\neg x \vee (z = 1)).$$

One satisfying assignment is

$$x = \text{True}, \quad y = 1, \quad z = 1.$$

This extension allows us to incorporate arithmetic relations and mixed logical–numerical constraints, testing whether representation-induced effects persist beyond purely Boolean structure.

We further extend this formulation to CSPs with Abelian group variables by replacing ordered arithmetic domains with algebraic domains endowed with group structure. Crucially, this extension increases the expressive richness of the constraint language while retaining precise control over logical equivalence: isomorphic instances preserve variable domains, group operations, and constraint structure, and therefore admit identical solution sets despite differences in representational form.

### 2.2. Isomorphic Paired Representations

Building on this shared logical core, we construct paired representations of each SAT instance in two surface forms: an abstract formal language (FL) representation and a concrete natural language (NL) representation. The former is instantiated as executable Z3 code with explicit variables and constraints, while the latter expresses the same constraints implicitly through natural language descriptions.

Prior work on constructing logic–language pairs typically follows one of two paradigms: template-based translations, which guarantee logical fidelity but severely limit linguistic and structural diversity, and large-language-model-based translations, which produce more natural and varied language but lack reliable mechanisms for verifying logical equivalence. This inherent trade-off makes existing datasets ill-suited for isolating representational effects under controlled semantic invariance.

To address this limitation, we develop a concretization framework. We begin by programmatically generating FL representations in Z3. We then apply a dual-learning-based translation framework (Xia et al., 2016), which couples two translators trained via reinforcement learning in opposite directions: one maps FL representations to NL descriptions with explicit variable bindings, and the other maps NL representations back to FL. An NL representation is retained only if the FL representation reconstructed from it is logically isomorphic to the original FL instance. Here, logical isomorphism means that the reconstructed representation encodes the same Boolean constraints up to a consistent renaming of variables.

To encourage the generation of more challenging NL instances, we further incorporate a difficulty-aware filtering procedure based on the pass rate of a reasoning model, preferentially retaining translations that induce lower solution success rates. Additional details of the dual-learning translation framework are provided in Appendix C.

All data used to train the translators and abstraction-aligned solvers are strictly separated from the evaluation sets. In our main experiments, translation-based training uses 5,000 instances, consisting of 1,000 instances from each of the

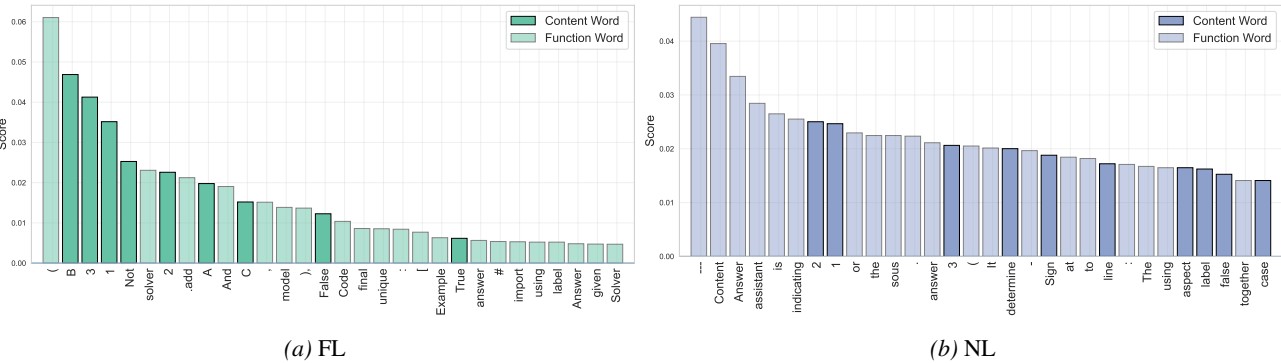

*Figure 1.* Top 30 tokens with the highest Grad × Input influence scores for Qwen3-30B-A3B on FL and NL representations.

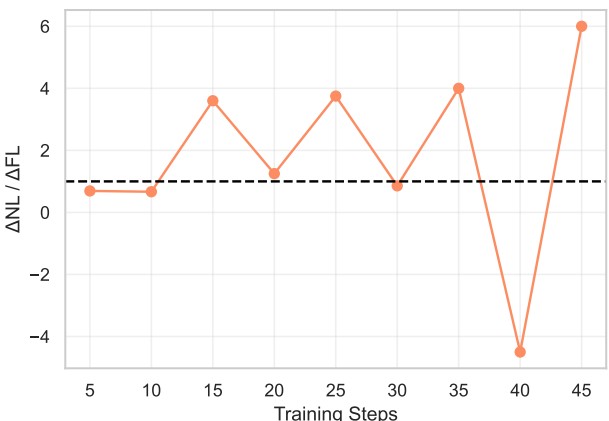

*Figure 2.* Learning transfer efficiency of Qwen3-30B-A3B when trained solely on FL data.

five problem configurations, with no instance overlap with any reported in-domain or out-of-domain evaluation benchmark. This separation is important because the evaluation is intended to measure cross-formulation reasoning rather than memorization of paired translations.

## 3. Concretization-Induced Failure Modes

Using the proposed concretization framework, we efficiently construct paired datasets of abstract FL representations and concrete NL representations that preserve identical underlying reasoning logic. In this work, we consider variable-size settings of $3 \times 3$, $3 \times 5$, and $5 \times 5$. For each setting, we collect 500 paired instances that satisfy a predefined difficulty criterion, defined as a pass rate below $8/16$ rollouts when solved by Qwen3-30B-A3B.

### 3.1. Performance Degradation after Concretization

We report the accuracy of state-of-the-art reasoning LLMs on both the original formal language (FL) templates and their corresponding natural language (NL) puzzles in Ta-

ble 1. As shown, nearly all models achieve high accuracy on the FL representations but exhibit substantial performance degradation after concretization into NL representations.

This degradation is most pronounced for Qwen3-30B-A3B, where the largest performance gap occurs in the $3 \times 3$ setting, with accuracy decreasing by 66%. For DeepSeek-R1 and Gemini-2.5-Pro, the most significant drops are observed on the $3 \times 5$ puzzles, with accuracy reductions of 37.4% and 24.79%, respectively. In contrast, GPT-o3 maintains relatively stable performance, though it still incurs a 2.4% accuracy decrease on the $3 \times 3$ puzzles.

To rule out the possibility that this gap is driven solely by narrative complexity, we also evaluate on the template-based K&K benchmark (Xie et al., 2024), whose NL constraints are direct, unambiguous statements without story-level distractors. The same qualitative pattern persists: Qwen3-30B-A3B and GPT-oss-20B show opposite FL/NL accuracy orderings, yet their FL and NL error sets remain nearly disjoint (Jaccard below 0.12). This rules out a simple difficulty account, under which errors on the harder formulation should largely contain or be contained by errors on the easier formulation. Full K&K accuracy, CKA, and attribution diagnostics are reported in Appendix G.1.

### 3.2. Token-Level Attribution Shift

To investigate why different representations yield divergent predictions, we measure the causal sensitivity of each input token using Grad × Input influence scores on the Qwen3-30B-A3B model. The objective function $J$ is defined as the total log-likelihood of the gold answer sequence, computed by summing the negative cross-entropy loss across the answer span. Gradients are enabled only for the prompt embeddings, and we backpropagate through $J$ to estimate token-level contributions. For each prompt token $i$, the Grad × Input influence score is defined as

$$\text{saliency}_i = \sum_{d=1}^{D} \frac{\partial J}{\partial e_{i,d}} e_{i,d}, \qquad (1)$$

where $\mathbf{e}_i = (e_{i,1}, \ldots, e_{i,D}) \in \mathbb{R}^D$ denotes the embedding vector of token $i$, $\frac{\partial J}{\partial e_{i,d}}$ is the gradient of $J$ with respect to the $d$-th component of $\mathbf{e}_i$, and $D$ is the embedding dimension. Finally, we apply L1 normalization across tokens to ensure comparability.

Figure 1 shows the top 30 tokens with the highest Grad×Input influence scores. In the FL representation, the most influential tokens are predominantly content-bearing elements, such as logical operators (e.g., Not, False) and variable identifiers (e.g., B3). In contrast, in the NL representation, the highest-influence tokens are frequently function words or structural artifacts, including template words (e.g., Content, is) and even non-semantic symbols (e.g., "—").

This pattern suggests that certain NL formulations may draw the model's attention toward tokens that are weakly related to the underlying logical structure, thereby reducing effective focus on reasoning-relevant information. We hypothesize that this effect reflects biases induced by training data distributions: tokens that commonly appear in non-reasoning contexts may prompt the model to rely less on explicit logical reasoning when processing such inputs.

### 3.3. Representation-Level Disparity

To examine whether the observed representation-dependent sensitivities correspond to deeper differences in internal representations, we further adopt linear Centered Kernel Alignment (CKA) (Kornblith et al., 2019). Given two instance-level representation matrices $X \in \mathbb{R}^{N \times d_1}$ and $Y \in \mathbb{R}^{N \times d_2}$, CKA is defined as

$$\text{CKA}(X, Y) = \frac{\|X_c^\top Y_c\|_F^2}{\|X_c^\top X_c\|_F \|Y_c^\top Y_c\|_F}, \qquad (2)$$

where $X_c$ and $Y_c$ denote representations centered across instances.

Figure 3 reports layer-to-layer linear CKA similarities between hidden states elicited by strictly isomorphic FL and NL inputs for three settings. In both FL vs. FL and NL vs. NL comparisons, we observe consistently high CKA values across most layer pairs, indicating strong internal representational coherence within each representation. In contrast, the FL vs. NL heatmap exhibits uniformly low CKA values across all layer pairs, with no pronounced diagonal or off-diagonal concentration. Notably, this lack of similarity persists even at deeper layers, where representations are often assumed to be more abstract or task-agnostic.

The sharp contrast between high within-representation similarity and low cross-representation similarity indicates that the reasoning model organizes the same set of problem instances into distinct representational geometries depending on surface form.

### 3.4. Weakly Coupled Learning Dynamics

Beyond behavioral discrepancies and representational separation, we further investigate whether FL and NL representations are coupled at the level of learning dynamics. We train the model exclusively on FL representations and track the evolution of both FL and NL accuracy throughout training. Since the model already attains high accuracy on FL, prolonged optimization would quickly lead to ceiling effects. We therefore restrict training to 45 optimization steps, during which FL accuracy of Qwen3-30B-A3B exceeds 98%. Checkpoints are saved every five steps, and both FL and NL accuracy are evaluated at each checkpoint. For each interval, we compute accuracy increments relative to the previous checkpoint to probe cross-representation learning behavior.

To quantify the degree of coupling between FL and NL learning, we analyze the incremental transfer efficiency, defined as the ratio between successive accuracy gains on NL and FL, $\Delta\text{NL}/\Delta\text{FL}$, across training steps. Figure 2 plots this ratio over the course of training. Rather than remaining stable, the ratio exhibits pronounced variability, including large fluctuations, sign changes, and occasional spikes when FL accuracy is already near saturation. This highly unstable trajectory suggests that improvements on NL are not smoothly governed by the same optimization signals that drive FL learning. Instead, NL gains arise irregularly and are highly sensitive to specific parameter updates, consistent with indirect or opportunistic transfer rather than coordinated learning within a shared reasoning process.

This weak coupling is not specific to the FL-to-NL direction. In a reverse experiment, we train exclusively on NL inputs and track FL accuracy over the same 45-step horizon. The resulting $\Delta\text{FL}/\Delta\text{NL}$ ratios again show large fluctuations and frequent sign changes, indicating that cross-representation transfer remains unstable even when optimization starts from the more concrete formulation. Detailed reverse-transfer statistics are provided in Appendix G.2.

## 4. Abstraction Alignment as Intervention

While the preceding analyses demonstrate that concretization from formal language (FL) to natural language (NL) representations can substantially degrade reasoning performance and alter internal computation, we therefore investigate abstraction alignment as an explicit intervention strategy. Our core intuition is that, if different surface representations induce divergent internal reasoning processes, then enforcing a shared abstraction layer, one that captures the underlying logical structure independent of surface form, may reduce representation-induced fragmentation.

We adopt translation-based alignment as a particularly direct and stringent instantiation of abstraction alignment. Specifically, we enhance the model's ability to translate

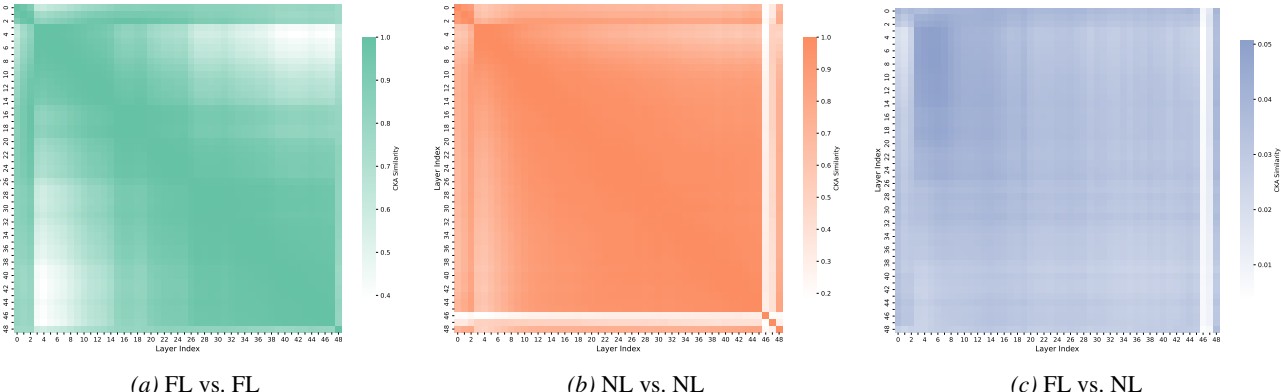

*(a)* FL vs. FL           *(b)* NL vs. NL           *(c)* FL vs. NL

*Figure 3.* Layer-to-layer linear CKA heatmaps of Qwen3-30B-A3B comparing instance-level representations.

a concretized NL formulation back into its corresponding FL representation. Translation requires the model to preserve functional equivalence across representations while discarding surface-specific features, thereby explicitly constraining it to operate on representation-invariant reasoning content rather than lexical or syntactic cues. As such, translation provides a strong and principled testbed for evaluating whether abstraction alone is sufficient to improve cross-representation reasoning performance.

### 4.1. Prompt-Based Abstraction Alignment

To mitigate the reasoning performance gap of LLMs when transitioning from FL representation to natural language puzzles, we propose a prompt-based method that encourages the model to extract the underlying reasoning logic before solving the task. Specifically, the solving model is first prompted to translate the natural language puzzle into a formal language template, and then to solve this formal representation in a second step to derive the final answer. To address the tendency of reasoning models to deviate from instructions, the prompt requires the LLM to explicitly output the reconstructed formal language template. The full prompt is provided in Appendix A.

As shown in Table 1, introducing the prompt-based reasoning-logic abstraction step leads to clear performance improvements for most LLMs across all three types of puzzles. Notably, Qwen3-30B-A3B achieves a 34% accuracy increase on the SAT problem with $3 \times 3$ variables. Another outlier is Gemini-2.5-Pro, which exhibits an 11.6% decrease in accuracy on the SAT problem with $5 \times 5$ variables when using the prompt-based method. A closer inspection of its outputs shows that Gemini-2.5-Pro often produces the final answer immediately after the translation step, neglecting the deeper symbolic reasoning required. This suggests that effective reasoning-logic abstraction must be paired with the ability to sustain coherent reasoning over the longer output sequences introduced by this process.

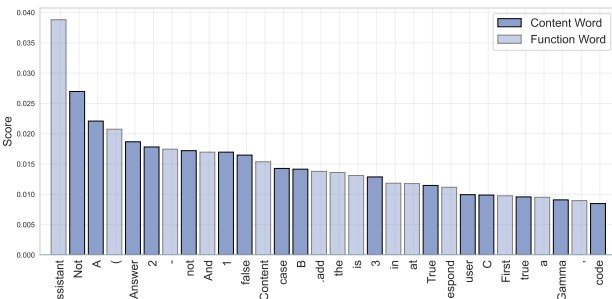

*Figure 4.* Top 30 tokens with the highest Grad $\times$ Input influence scores for Qwen3-30B-A3B after abstraction-alignment training.

### 4.2. RL-Based Abstraction Alignment

To further strengthen the model's ability to perform reasoning-logic abstraction, we introduce a complementary training-based approach. Specifically, the solving model is trained via reinforcement learning to translate NL representations back into their corresponding FL representations. The reward depends solely on whether the generated FL representation is isomorphic to the original one, thereby encouraging the model to map diverse surface formulations to a shared underlying logical structure.

When this abstraction ability is strengthened through training, the performance of both Qwen3-30B-A3B and GPT-oss-20B improves further, approaching their respective accuracy levels on the original formal language templates. Notably, Qwen3-30B-A3B on the SAT problem with $3 \times 3$ variables, as well as GPT-oss-20B across all SAT settings, even surpass their performance on the formal templates. These results indicate that abstraction-enhanced training can reduce the dispersion of LLM reasoning across symbolic and natural-language formulations, enabling more consistent and effective reasoning across representations.

We also compare against a generic abstraction-before-solving baseline inspired by Abstraction-of-Thought (AoT),

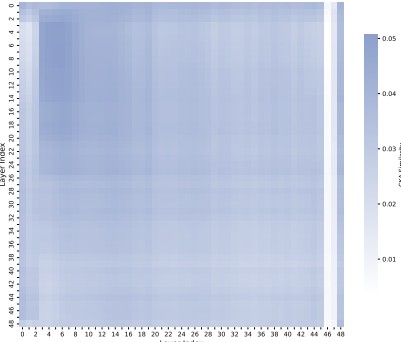

*Figure 5.* Layer-to-layer linear CKA heatmaps of Qwen3-30B-A3B comparing FL and NL representations after abstraction-alignment training.

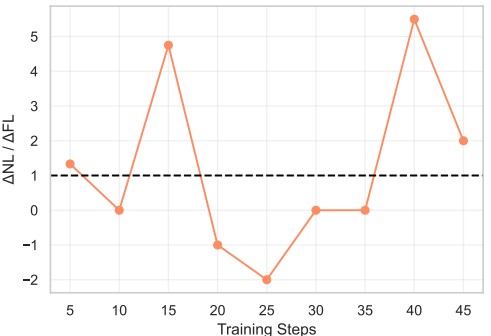

*Figure 6.* Learning transfer efficiency of abstraction-aligned Qwen3-30B-A3B when trained solely on FL data.

which encourages intermediate abstraction but does not require an explicit FL reconstruction. On Qwen3-30B-A3B, this baseline improves the Bool average from 28.2 to only 30.4, far below the prompt-based result of 51.6 and the RL-based result of 75.2. This suggests that the gain is not simply due to inserting an intermediate reasoning step or increasing output length; effective abstraction requires either explicit structural reconstruction or a verifiable isomorphism reward.

### 4.3. Out-of-Domain Generalization

We further evaluate the abstraction-enhanced model on several publicly available benchmarks from prior work (Valmeekam et al., 2023; Zheng et al., 2024). For PlanBench, we focus specifically on the plan-generation task. As shown in Table 2, abstraction-enhanced training substantially improves robustness to perturbations such as injected typos, irrelevant statements, adversarial typographic changes, and adversarial triggers. Moreover, both models also achieve higher performance on real-world planning and mathematical reasoning tasks, with the abstraction-enhanced GPT-oss-20B showing an 8.2% improvement on PlanBench and a 9.0-point gain on perturbed CatAttack. These results highlight abstraction alignment training as an effective strategy for enhancing the robustness of LLM reasoning beyond our synthetic construction pipeline.

## 5. Alignment Effectiveness

We next analyze how abstraction alignment changes the failure modes identified in Section 3. The structure mirrors the earlier diagnostics: Section 5.1 revisits token attribution, Section 5.2 revisits representation geometry, Section 5.3 revisits learning dynamics, and Section 5.4 uses activation patching to test whether the observed gains are mediated by causal routing rather than representational fusion.

### 5.1. Attribution Repair Analysis

After training for abstraction ability, we observe a systematic shift in token-level Grad × Input influence scores on the NL representation. As shown in Figure 4, the most influential tokens for the Qwen3-30B-A3B model transition from predominantly function words to content-bearing tokens that directly encode logical entities, relations, and constraints, aligning closely with the attribution patterns observed in the FL representation.

This attribution repair indicates that translation-based abstraction training effectively redirects the model's attention toward reasoning-relevant components of the input, mitigating earlier failure modes in which surface-level linguistic artifacts dominated the decision process.

### 5.2. Persistent Representation-Level Disparity

Despite the clear behavioral and attribution-level improvements induced by translation-based abstraction training, representation-level alignment remains strikingly limited. Figure 5 reports layer-to-layer linear CKA similarities between hidden states elicited by FL and NL inputs in abstraction-aligned models.

Across nearly all layers, cross-representation similarity remains low and diffuse, with no clear diagonal structure emerging after alignment. This indicates that corresponding layers do not converge to shared representational subspaces, even when both inputs encode identical logical constraints and the model has been trained with explicit abstraction alignment objectives.

### 5.3. Persistent Decoupling in Learning Dynamics

Figure 6 plots this ratio over training steps for the abstraction-aligned model. Strikingly, the resulting trajectory closely resembles that observed prior to abstraction alignment. The transfer efficiency continues to exhibit pronounced variability, including large fluctuations, frequent

| Model | Method | Natural-Plan | | | PlanBench | R2ATA Clean/Pert. | CatAttack Clean/Pert. |
|---|---|---|---|---|---|---|---|
| | | Calendar | Meeting | Trip | | | |
| **Qwen3-30B-A3B** | Orig. | 84.80 | 12.30 | 3.75 | 68.20 | 90.47/86.87 | 96.50/94.50 |
| | RL | 86.20 (+1.40) | 14.10 (+1.80) | 4.94 (+1.19) | 73.20 (+5.00) | 91.18/87.50 | 96.50/96.00 |
| **GPT-oss-20B** | Orig. | 83.90 | 4.00 | 0.00 | 47.40 | 79.18/70.81 | 63.00/61.00 |
| | RL | 85.70 (+1.80) | 9.60 (+5.60) | 0.06 (+0.06) | 55.60 (+8.20) | 82.41/76.11 | 70.60/70.00 |

*Table 2.* Out-of-domain performance of Qwen3-30B-A3B and GPT-oss-20B before (Orig.) and after training-based abstraction alignment (RL). R2ATA reports adversarial typographic perturbations on GSM8K, BBH, and MMLU; CatAttack reports adversarial triggers on NuminaMath. Prompt-based abstraction is not included here because these benchmarks do not provide paired FL ground truth for explicit reconstruction.

sign changes, and occasional spikes when FL accuracy is already near saturation.

This persistent instability indicates that abstraction alignment does not induce a tightly coupled optimization regime between FL and NL representations. Despite the substantial behavioral improvements and attribution-level repairs observed after alignment, learning on FL still does not translate into smooth, proportional improvements on NL. Instead, cross-representation gains remain sporadic and highly sensitive to individual parameter updates.

### 5.4. Representation-Conditional Routing Channels

While previous analyses show that translation-based training substantially improves cross-representation performance without collapsing representational geometry, the mechanism by which such improvements arise remains unclear.

To investigate this question, we analyze routing specificity using layer-wise activation patching. Specifically, for an NL input $x^{\mathrm{NL}}$ and its isomorphic FL counterpart $x^{\mathrm{FL}}$, we replace the hidden states at layer $\ell$ in the NL forward pass with the corresponding hidden states from the FL forward pass and continue the computation. Let $y^{\star}$ denote the gold answer and $\mathcal{Y}^{-}$ the set of incorrect candidate answers. We define

$$
\begin{aligned}
\Delta_{\mathrm{gold}}^{(\ell)} = {} & \log p_{\mathrm{patch}}(y^{\star} \mid x^{\mathrm{NL}}) \\
& - \log p_{\mathrm{orig}}(y^{\star} \mid x^{\mathrm{NL}}),
\end{aligned}
\tag{3}
$$

and

$$
\begin{aligned}
\Delta_{\mathrm{wrong}}^{(\ell)} = {} & \frac{1}{|\mathcal{Y}^{-}|} \sum_{y \in \mathcal{Y}^{-}} \Big[ \log p_{\mathrm{patch}}(y \mid x^{\mathrm{NL}}) \\
& - \log p_{\mathrm{orig}}(y \mid x^{\mathrm{NL}}) \Big].
\end{aligned}
\tag{4}
$$

Routing specificity is $\Delta_{\mathrm{gold}}^{(\ell)} - \Delta_{\mathrm{wrong}}^{(\ell)}$. A matched patch uses FL hidden states from the isomorphic instance with the same solution, while a mismatched patch uses hidden states from a different instance. Thus, higher matched specificity

indicates that the intervention transfers content-specific reasoning information rather than generic properties of FL inputs.

Figure 7a reports the resulting routing specificity under matched and mismatched representations, both before and after training. We observe that routing specificity consistently peaks in the same mid–upper layers (approximately layers 38–44) for both the original and abstraction-aligned models. Importantly, translation-based training does not shift the locations of these peaks, but systematically amplifies their magnitude. This indicates that the layer-wise routing structure is already present prior to training and is strengthened rather than newly created by abstraction alignment.

Moreover, matched representations exhibit substantially higher routing specificity than mismatched ones across all layers, both before and after training. This asymmetry suggests that the model selectively routes information along representation-consistent computational paths, rather than uniformly amplifying logits or activation magnitudes. Training increases this selectivity without inducing representational alignment, consistent with a routing-based mechanism rather than representational fusion.

To further isolate this effect, Figure 7b plots the routing gap, defined as the difference between routing specificity under matched and mismatched representations. We find that translation-based training consistently enlarges this gap across the same mid–upper layers, indicating enhanced conditional routing selectivity.

## 6. Related Work

### 6.1. Formulation Sensitivity of LLMs

Since the advent of LLMs, prior studies have shown extreme sensitivity of LLMs to input formulation. For instance, (Errica et al., 2025) and (He et al., 2024) demonstrate that input formatting alters results, while (Ackerman et al., 2024) and (Qiang et al., 2024) highlight the impact

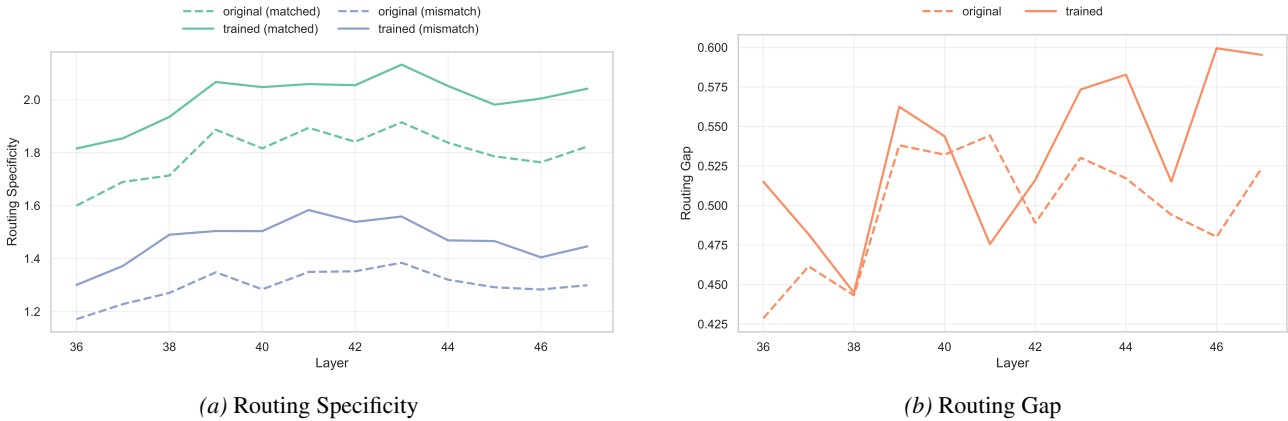

*(a)* Routing Specificity

*(b)* Routing Gap

*Figure 7.* Routing specificity analysis for the original and abstraction-aligned Qwen3-30B-A3B model.

of synonymous paraphrases. Similarly, (Gan et al., 2024) show that replacing critical tokens with predefined typos from a common misspelling dictionary can alter outcomes. (Zhou et al., 2024) further demonstrate that paraphrasing questions through prompt-based methods also affects performance. In addition, both (Zhu et al., 2024) and (Hu et al., 2025) show that simply changing the input language can influence results, while (Rajeev et al., 2025) and (Yang et al., 2025b) demonstrate that introducing irrelevant information can similarly degrade performance. These works show that heuristic modifications to prompts often hurt LLM reasoning on benchmarks. However, these studies typically assume that the underlying reasoning process stays the same, since the changes are defined as "meaning-preserving." Critically, this assumption is rarely supported by rigorous verification of whether input–output reasoning consistency is actually maintained after such perturbations.

To address this gap, (Fu et al., 2024) propose training a smaller model to align input formulations with LLM preferences, while (Zhao et al., 2024) enhance robustness by augmenting supervised fine-tuning data with perturbed variants to enforce output consistency. However, both approaches operate primarily at the surface-text level, without explicitly teaching LLMs to model the reasoning logic that should remain invariant across semantically equivalent formulations.

### 6.2. Translation from Formal to Natural Language

As high-level abstractions of real-world tasks, much prior work has focused on instantiating formal language skeletons into natural language puzzles that fit practical scenarios. For example, (Kazemi et al., 2023) propose BoardgameQA, which maps board game rules into natural language QA, emphasizing contradictory information and preference reasoning. (Lin et al., 2025) formalize logic grid and zebra puzzles as CSPs. (Wei et al., 2025) generate narrative logic puzzles automatically from SAT formulas. (Sinha et al., 2019)

transform kinship rules into short stories with associated QA. While these enrich benchmarks, they rely heavily on human quality control and focus on reasoning consistency, overlooking semantic difficulty. As a result, benchmarks emphasize reasoning steps, whereas real-world challenges often lie in mapping complex contexts into abstract logic.

### 6.3. Abstraction-Based Reasoning

Several recent methods encourage LLMs to reason through abstract intermediate structures, including Chain-of-Abstraction (CoA), Abstraction-of-Thought (AoT), and AbstRaL. Our goal differs in two respects. First, we use abstraction not only as an accuracy-improving strategy, but also as an intervention for probing whether FL and NL reasoning pathways become representationally fused or remain compartmentalized. Second, our training signal directly supervises abstraction quality through logical isomorphism verification, whereas generic intermediate-abstraction methods can be satisfied by underspecified placeholders or answer-correctness rewards that do not require reconstructing the underlying constraint structure. The AoT-style baseline in Section 4 confirms this distinction empirically: implicit abstraction yields only marginal gains unless the model is required, by prompting or reward, to recover a verifiable formal structure.

## 7. Conclusion

Using strictly isomorphic FL–NL tasks, we show that a substantial portion of formulation sensitivity in LLM reasoning arises from compartmentalized reasoning. FL and NL inputs induce distinct reasoning behaviors, weakly coupled learning dynamics, and persistently separated representations. Abstraction training mitigates these failures without collapsing representational geometry. Instead, it selectively amplifies pre-existing, representation-conditioned routing pathways that regulate access to internal reasoning.

## Acknowledgements

Supported by Shanghai Artificial Intelligence Laboratory.

## Impact Statement

This work aims to improve the reliability and interpretability of large language model reasoning by identifying when logically equivalent inputs induce separated internal computation. The proposed abstraction-alignment approach may help make reasoning systems more robust to changes in formulation, which is beneficial for applications where users express the same problem in different surface forms. At the same time, improved reasoning robustness can also strengthen systems used in high-stakes or adversarial settings, so deployment should be accompanied by task-specific validation, monitoring, and safeguards against over-reliance on model outputs. Our experiments use synthetic and public benchmarks and do not involve human-subject data.

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

# A. Prompt

---

**Formal Language Template Prompt**

Code:
```
from z3 import *
solver = Solver()
A1, A2, A3, B1, B2, B3, C1, C2, C3 = Bools('A1 A2 A3 B1 B2 B3 C1 C2 C3')
solver.add(Not(And(Not(A1), Not(A2))))
...
solver.add(Not(And(A1, B2)))
solver.add(Not(And(A1, Not(A3))))
```

Determine the truth value (True or False) for each variable defined in the given Code.
Respond with your final answer using the label "Final Answer". Format each line as: "[Variable name]: [True/False]".

Example:
Final Answer:
A1: True
B2: False

---

**Natural Language Puzzle Prompt**

Content:
In the once-thriving Kingdom of the Mages, the great dragons were both guardians andiphers of ancient mystery. Among these dragons was one known as Ember, who guarded the last remnants of the royal lineage and the treasures that lay beneath the crumbling towers of the ancient castle. For centuries, Ember had ...
Ember, with her uncanny ability to discern the true intentions of those who dared to challenge her, began to probe Arin's resolve. Their encounter ...
Here are the constraints that governed his plan:
**The First Challenge**: Ember's gaze locked onto Arin's, her emerald eyes ...
...
**The Ninth Challenge**: Ember's voice took on a tone of finality as she delivered ...
Ember's words hung in the air, a testament to the intricate web of conditions that bound Arin's quest. The warrior knew that his success depended not only on his own courage but also on the willingness of others to support his cause. As he prepared to face the dragon, he understood that his journey was not just one of sword and fire but of logic, resolve, and the ability to navigate a labyrinth of interdependent choices.

Definitions:
A1: Arin must slay the dragon to achieve his goal.
A2: Arin must attain the throne to fulfill his purpose.
...
C2: The nobles must not oppose Arin for his rule to be secure.
C3: The people must know peace for Arin's quest to be truly successful.

Based on the Content and Definitions, determine the truth value (True or False) for each variable mentioned.
Respond with your final answer using the label "Final Answer". Format each line as: "[Variable name]: [True/False]".
Each variable name appears at the start of its corresponding definition in the Definitions.

Example:
Final Answer:
A1: True
B2: False

---

---

**Natural Language Puzzle Prompt with Back Translation**

Content:
Adam sat on the cold mountainside, lying on the soft peat, a thin reed sticking into his back. The rain pelted him ...
Here are the constraints that governed his plan:
Either Adam remembered to pack his fireproof container or he remembered to bring his emergency flares, both could not be forgotten at the same time.
...
If Adam didn't remember to pack his fireproof container, then the encryption key wasn't secure.

Definitions:
A1: Adam remembered to pack his fireproof container.
...
C5: The final encryption key was in place.

Based on the Content and Definitions, determine the truth value (True or False) for each variable mentioned. First, Convert the Content into Z3 code. Each constraint should represent a forbidden combination of assignments for two variables. Then, Solve the Z3 code to obtain the final truth values.
Respond with the translated Z3 code, labeled as "Final Z3 Code:" and provide the final answers using the label "Final Answer:". Format each line in final answer as: "[Variable name]: [True/False]". Each variable name appears at the start of its corresponding definition in the Definitions.

Example:
Final Z3 Code:
from z3 import *
solver = Solver()
A1, A2, A3, B1, B2, B3 = Bools('A1 A2 A3 B1 B2 B3')
solver.add(Not(And(Not(A2), Not(B1))))
...
solver.add(Not(And(Not(A1), B2)))

Final Answer:
A1: False
...
B3: True

---

## Translation Formal Language Template to Natural Language Puzzle

Code:
from z3 import *
solver = Solver()
A1, A2, A3, B1, B2, B3, C1, C2, C3 = Bools('A1 A2 A3 B1 B2 B3 C1 C2 C3')
solver.add(Not(And(Not(A1), Not(A2))))
...
solver.add(Not(And(A1, Not(A3))))

Background:
So many times have I walked on ruins, the remainings of places that I loved and got used to.. At first I was scared, each time I could feel my city, my current generation collapse ...

Integrate all information from the Z3 code into the Background to generate a challenging natural language content. Do not refer to or quote the code directly, and do not use symbolic identifiers (e.g., "A1", "C5") in the narrative. Ensure that each constraint encoded in the Z3 code is explicitly represented in the final version of the natural language content, each constraint should be clearly reflected one by one, while the final solution must remain undisclosed. After that, provide natural language definitions for each variable used in the code. Each line formatted as: "[Variable name]: [Definition in the natural language content]".

Conclude your response with following format:
Natural Language Content:
[content]

Definitions:
[definitions]

## Translation Natural Language Puzzle to Formal Language Template

Content:
The story of "The Really Bad Decision" is a cautionary tale of hubris, miscommunication, and the consequences of half-hearted efforts. At its core, it is a narrative of ...
**Not(And(Not(A2), Not(B1)))**: This constraint prohibits the simultaneous absence of A2 and B1...
...
**Not(And(A3, Not(C2)))**: This constraint ensures that A3 and C2 cannot both be present and absent...

Definitions:
A1: Represents the implementation of a critical initial design review or feasibility study.
...
C3: Represents the implementation of a fail-safe mechanism.

Based on the Definitions, translate the Natural Language Content into Z3 code. Each constraint consists of a forbidden combination of assignments for two variables.
Conclude your response with "Final Z3 Code:". Then present the generated code directly, do not enclose it in quotation marks or code blocks.

For example:
Final Z3 Code:
from z3 import *
solver = Solver()
A1, A2, A3, B1, B2, B3 = Bools('A1 A2 A3 B1 B2 B3')
solver.add(Not(And(Not(A2), Not(B1))))
...
solver.add(Not(And(Not(A1), B2)))

# B. Algorithms

---

**Algorithm 1** Generate SAT Template

---

**Input**: rows $M$, cols $N$
**Output**: Set $Constraints$ such that the SAT instance has a *unique* model

1: Initialize variables: $Vars \leftarrow \{A_1, A_2, \ldots, A_{M \times N}\}$
2: $Constraints \leftarrow \emptyset$
3: Initialize incremental SAT solver $S$      // empty constraint stack
4: $S$.PUSH()      // level-0 frame
5: **loop**
6:     **if** $S$.CHECK() = UNSAT **then**
7:       $S$.POP()      // remove last constraint
8:       Remove last constraint $last\_c$ from $Constraints$
9:       $model \leftarrow S$.MODEL()      // solver is SAT again
10:      $found \leftarrow$ **false**
11:      **for all** distinct pairs $(v_i, v_j)$ in $Vars$ **do**
12:        $c\_cand \leftarrow \neg(v_i = model[v_i] \wedge v_j = model[v_j])$
13:        $S$.PUSH(); $S$.ADD($c\_cand$)
14:        **if** $S$.CHECK() = SAT **then**
15:          $found \leftarrow$ **true**
16:          $model \leftarrow S$.MODEL()      // new model
17:          Add $c\_cand$ to $Constraints$
18:          **break** the for-loop
19:        **else**
20:          $S$.POP()      // discard $c\_cand$
21:        **end if**
22:      **end for**
23:      **if not** $found$ **then**
24:        **return** $Constraints$      // unique model achieved
25:      **end if**
26:     **else**
27:       $model \leftarrow S$.MODEL()
28:       Randomly pick distinct $v_1, v_2 \in Vars$
29:       $c \leftarrow \neg(v_1 = model[v_1] \wedge v_2 = model[v_2])$
30:       $S$.ADD($c$); $Constraints \mathrel{+}= c$      // stay in same frame
31:     **end if**
32: **end loop**

---

---

**Algorithm 2** SAT Isomorphic

---

**Input**: $src\_code$, $tgt\_code$
**Output**: **True** if two SAT templates are isomorphic, **False** otherwise

1: $ns_{\text{src}} \leftarrow$ new namespace with Z3 pre-imported
2: Execute $src\_code$ in $ns_{\text{src}}$
3: $solver_{\text{src}} \leftarrow$ last $v$ in $ns_{\text{src}}$.VALUES() where $v$ is a Z3 solver
4: $A_{\text{src}} \leftarrow$ list of $solver_{\text{src}}$.ASSERTIONS()
5: $ns_{\text{tgt}} \leftarrow$ new namespace with Z3 pre-imported
6: Execute $tgt\_code$ in $ns_{\text{tgt}}$
7: $solver_{\text{tgt}} \leftarrow$ last $v$ in $ns_{\text{tgt}}$.VALUES() where $v$ is a Z3 solver
8: $A_{\text{tgt}} \leftarrow$ list of $solver_{\text{tgt}}$.ASSERTIONS()
9: $C_{\text{src}} \leftarrow \{ \_canonical(c) \mid c \in A_{\text{src}} \}$      // NNF + simplify + sort
10: $C_{\text{tgt}} \leftarrow \{ \_canonical(c) \mid c \in A_{\text{tgt}} \}$
11: **return** $C_{\text{src}} = C_{\text{tgt}}$

---

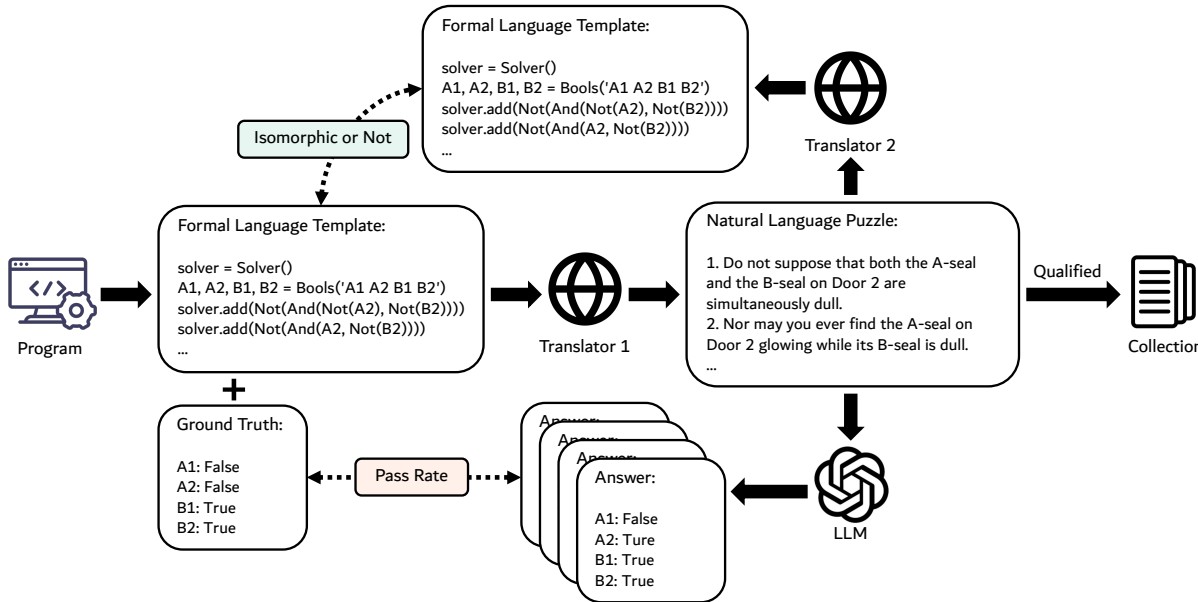

*Figure 8.* The construction process of a paired formal language template and natural language puzzle proceeds. First, a rule-based program generates a formal language template and its ground-truth assignment. This template is then passed to a translator, which converts it into a natural language puzzle. The puzzle is subsequently back-translated into a formal template by another translator and presented to an LLM, which produces multiple responses. A natural language puzzle is retained and collected if it passes isomorphism verification and its pass rate falls below a difficulty threshold.

## C. Dual-learning Translation Framework

### C.1. Inference Pipeline

Our concretization framework adopts the standard dual learning approach (Xia et al., 2016), which consists of two training cycles involving two translators. In the first cycle, Translator 1 translates a formal language template into a natural language puzzle, while Translator 2 translates the resulting puzzle back into a formal language template. Translator 1 serves as the optimization target in this cycle. In the second cycle, Translator 2 translates a real-world puzzle into a formal language template, and Translator 1 then translates the template back into a natural language puzzle. In this cycle, Translator 2 is the optimization target. The overall training process is illustrated in Figure 9.

### C.2. Dual-learning Reinforcement Learning

For Translator 1, the input is a constructed formal language template, and the output is a natural language puzzle together with variable definitions. The reward is derived from two components: (i) the pass rate of an answer model on the generated natural language puzzle, and (ii) the isomorphism decision between the original formal language template and the back-translated template produced by Translator 2. For Translator 2, the input is a real-world puzzle, and the output is a formal language template along with variable definitions. Its reward combines (i) a format check on the generated formal language template and (ii) the similarity between the original real-world puzzle and the natural language puzzle generated by Translator 1.

Through iterative training, our dual-learning framework converges toward a state where formal language templates can be automatically translated into natural language puzzles

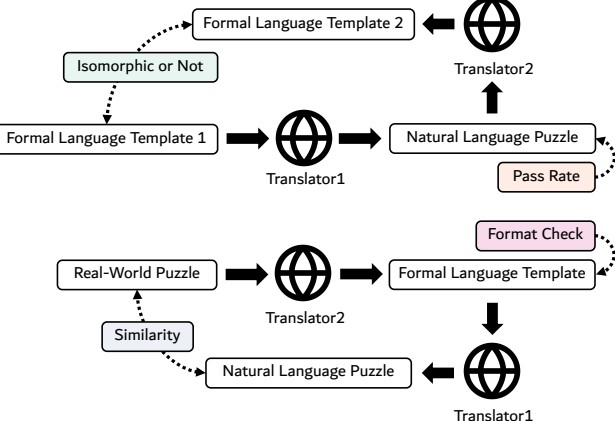

*Figure 9.* The training process for the natural language translator.

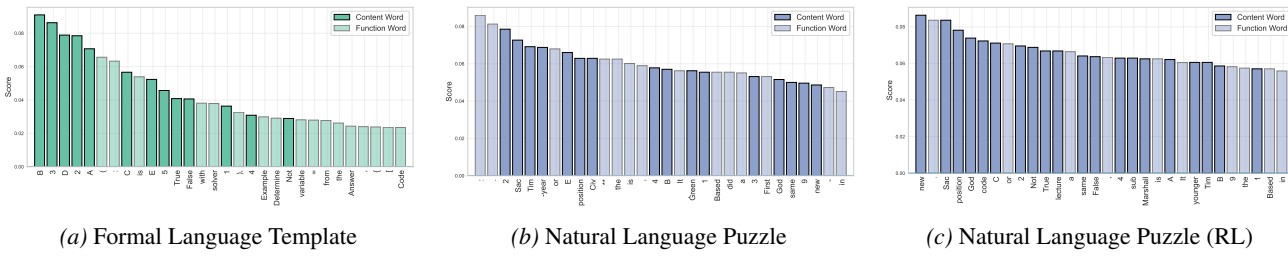

*(a)* Formal Language Template          *(b)* Natural Language Puzzle          *(c)* Natural Language Puzzle (RL)

*Figure 10.* Top 30 tokens with the highest Grad $\times$ Input influence scores.

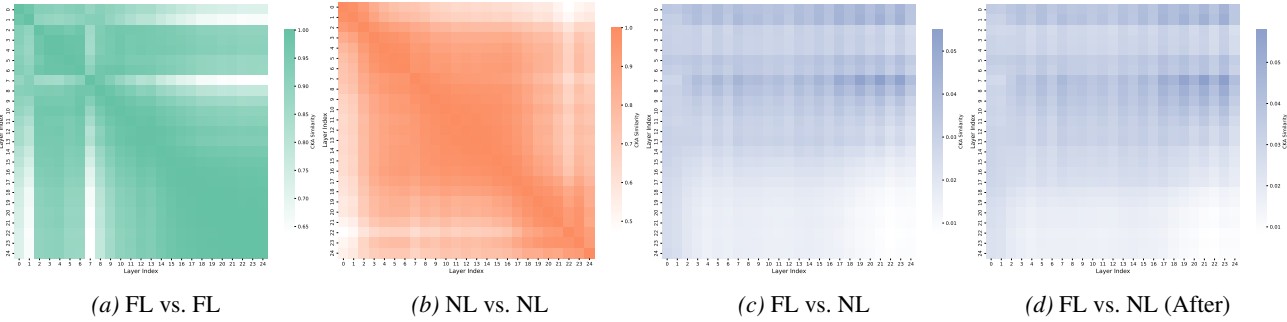

*(a)* FL vs. FL          *(b)* NL vs. NL          *(c)* FL vs. NL          *(d)* FL vs. NL (After)

*Figure 11.* Layer-to-layer linear CKA heatmaps comparing instance-level representations across input representations.

that are both challenging in formulation and logically consistent with the original formal representation.

### C.3. Implementation

In our translator implementation, we leverage the training set from the well-known logic puzzle benchmark *Knights-and-Knaves* (Xie et al., 2024) as the source of real-world puzzles and employ the Qwen3-30B-A3B model (Yang et al., 2025a) as the LLM solver. To verify the isomorphism between Formal Language Template 1 and the back-translated Formal Language Template 2, we apply Algorithm 2. For measuring the similarity between the Real-World Puzzle and the back-translated Natural Language Puzzle, we compute the BLEU score using the Qwen3-30B-A3B tokenizer. We extend the `verl` framework (Sheng et al., 2025) to enable the training of two translators based on the r1-distill-Qwen-32B model (Guo et al., 2025), each equipped with an independent LoRA adapter (Hu et al., 2022). Training is performed using the GRPO algorithm (Guo et al., 2025) and the AdamW optimizer (Loshchilov & Hutter, 2019). The two translators are trained alternately on two 8-card H800 GPU nodes with a learning rate of $1 \times 10^{-6}$. For decoding, we configure the parameters as follows: temperature = 1.0, top-p = 1.0, and LoRA rank = 8.

## D. Additional Result on GPT-oss-20B

### D.1. Input Token Influence

We also present the top 30 tokens with the highest Grad $\times$ Input influence scores for GPT-oss-20B. As shown in Figure 10, GPT-oss-20B exhibits a trend similar to Qwen3-30B-A3B: in both the formal-language template and the high-pass-rate natural-language puzzle settings, the most influential tokens are typically content words. In contrast, in the low-pass-rate natural-language puzzle, the tokens with the highest influence scores are often function words. After training for abstraction ability, the low-pass-rate natural language puzzle, the most influential tokens for the GPT-oss-20B shift from function words to content words as well.

### D.2. Layer-to-layer Linear CKA Heatmaps

As shown in Figure 11, consistent with our observations on Qwen3-30B, GPT-oss-20B exhibits high within-representation similarity in both the FL vs. FL and NL vs. NL settings, characterized by clear diagonal structure across most layers. In contrast, cross-representation similarity between FL and NL remains uniformly low across nearly all layers prior to alignment. After translation-based alignment, we observe only marginal increases in cross-representation CKA, with no

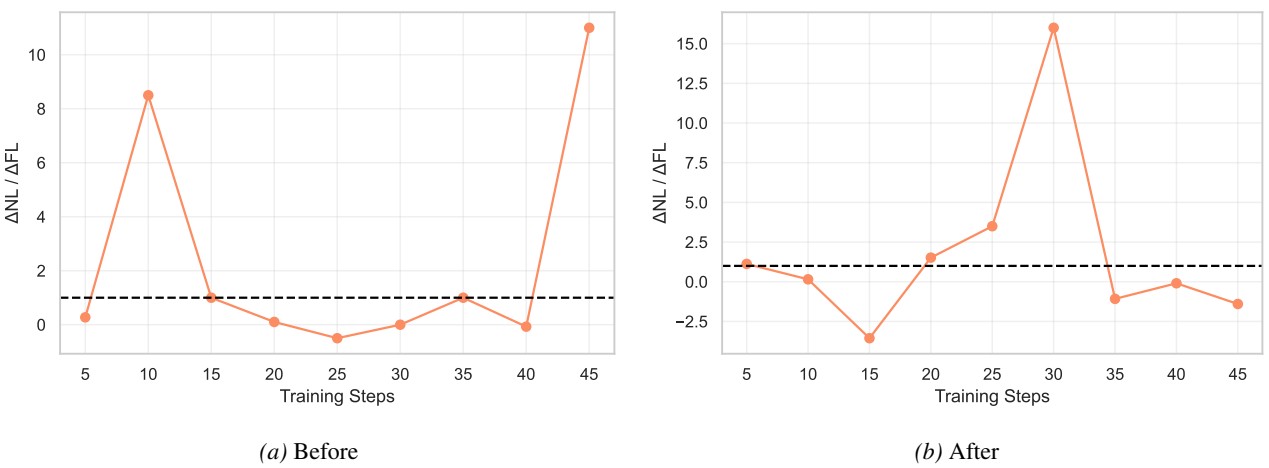

*(a)* Before

*(b)* After

*Figure 12.* Transfer Efficiency

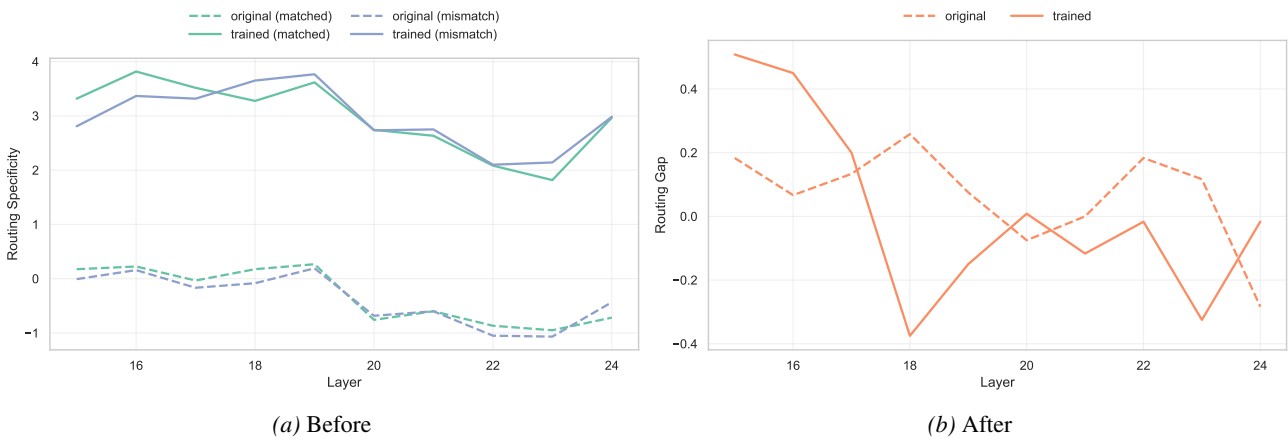

*(a)* Before

*(b)* After

*Figure 13.* Routing diagnostics for GPT-oss-20B.

emergence of a pronounced diagonal pattern.

Together, these results demonstrate that the persistence of representation-level separation between FL and NL is robust across model architectures, indicating that translation-based alignment improves cross-representation reasoning without inducing representational fusion.

### D.3. Learning Isolation

Figure 12 reports the incremental transfer efficiency $\Delta$NL/$\Delta$FL during FL-only training, before and after translation-based alignment. Similar to Qwen3-30B, the transfer efficiency trajectory exhibits substantial variability across training steps, including large fluctuations, sign changes, and occasional spikes. Importantly, this behavior persists after alignment: although translation-based training increases the overall magnitude of NL gains, the ratio between NL and FL improvements remains highly unstable and non-stationary. The absence of a stable or proportional relationship between $\Delta$NL and $\Delta$FL indicates that, as in larger models, cross-representation learning in GPT-OSS-20B does not follow a systematically coupled optimization process. Instead, NL improvements arise intermittently, consistent with weak or indirect transfer rather than shared learning dynamics.

### D.4. Routing Channels

As shown in Figure 13a, before alignment, both matched and mismatched representations exhibit routing specificity values close to zero across layers, with minimal separation between the two conditions. This indicates that representation-

conditioned routing is largely absent in the original model. After alignment, routing specificity increases substantially across a broad range of layers for both matched and mismatched settings, demonstrating that translation-based training amplifies the causal impact of patched activations. However, the two curves remain closely aligned, showing little systematic separation between matched and mismatched representations. Consistent with this observation, the routing gap in Figure 13b remains small and highly variable across layers, frequently changing sign. This suggests that, although alignment strengthens overall routing sensitivity in GPT-oss-20B, it does not induce stable representation-specific routing channels at this scale.

## E. Natural Language Puzzle Formulation Diversity

Beyond the challenge of formulation, our translation framework from formal language templates to natural language puzzles also demonstrates greater diversity compared to template-based methods. Specifically, we embed the formal language templates, the widely used natural language puzzle benchmark Knights-and-Knaves (Xie et al., 2024), and our constructed natural language puzzles using the Qwen3-30B-A3B tokenizer. The resulting embeddings are then projected into two dimensions using Principal Component Analysis (PCA).

As shown in Figure 14, both the formal language templates and Knights-and-Knaves puzzles exhibit concentrated distributions within relatively small regions. In contrast, our generated natural language puzzles display a far more dispersed distribution, suggesting that our translation framework effectively captures a broader and more diverse range of input formulations.

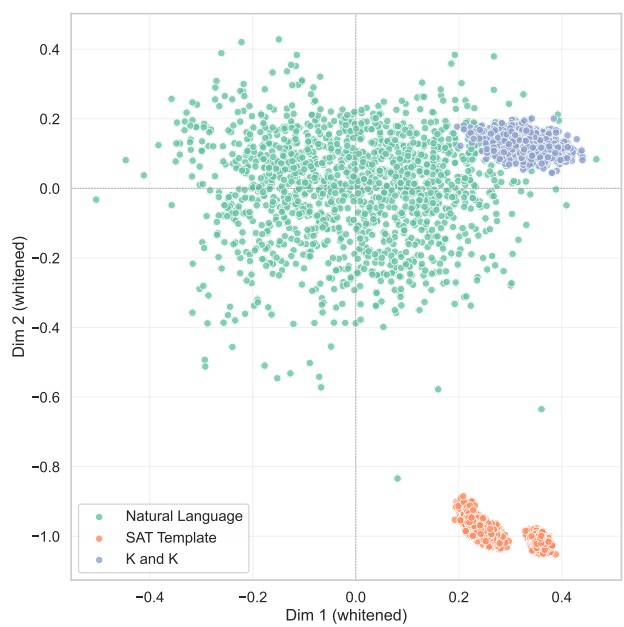

*Figure 14.* The embedding distribution comparison, reduced to two dimensions using PCA.

## F. Limitations

Our controlled SAT/CSP setting enables exact verification of FL–NL isomorphism and causal mechanistic analysis, but it does not cover all forms of mathematical or commonsense reasoning. Fully verified FL–NL equivalence is substantially harder to guarantee for open-ended mathematical proofs or long-form reasoning tasks, and extending the construction pipeline to those domains remains an important direction. In addition, the most detailed activation-patching and learning-dynamics analyses focus on Qwen3-30B-A3B and GPT-oss-20B because they require repeated checkpointing and hidden-state interventions. Our broader model and out-of-domain results suggest that the phenomenon is not model-specific, but full mechanistic replication across additional model families would strengthen the conclusion further.

## G. Additional Experiments

### G.1. Template-Based K&K Control

We evaluate on a public Knights-and-Knaves (K&K) benchmark (Xie et al., 2024) as an independent control for narrative complexity and construction bias. Unlike our generated puzzles, K&K uses fixed templates with direct constraint statements, involves no difficulty-aware filtering, and is produced by a translation pipeline independent of ours. Despite these differences, the same compartmentalization signatures persist.

| Model | FL Acc. | NL Acc. | Error Jaccard |
|---|---|---|---|
| Qwen3-30B-A3B (Orig.) | 85.43 | 93.29 | 0.0643 |
| GPT-oss-20B (Orig.) | 83.57 | 74.43 | 0.0884 |
| Qwen3-30B-A3B (RL) | 89.29 | 93.71 | 0.1098 |
| GPT-oss-20B (RL) | 87.86 | 80.14 | 0.1164 |

*Table 3.* K&K accuracy and error-set overlap. Opposite FL/NL accuracy orderings with near-disjoint error sets are inconsistent with a simple difficulty-only explanation.

| Model | CKA | ppl=5 | ppl=6 | ppl=7 | ppl=8 |
|---|---|---|---|---|---|
| Qwen3 (Orig.) | FL-FL | 89.44 | 87.53 | 87.70 | 86.77 |
|  | NL-NL | 82.89 | 81.83 | 82.18 | 81.71 |
|  | FL-NL | 62.08 | 54.79 | 53.00 | 51.66 |
| GPT-oss (Orig.) | FL-FL | 81.75 | 77.28 | 77.96 | 79.38 |
|  | NL-NL | 75.80 | 76.03 | 75.69 | 77.17 |
|  | FL-NL | 54.58 | 49.46 | 48.70 | 49.68 |
| Qwen3 (RL) | FL-FL | 89.56 | 87.55 | 87.82 | 86.79 |
|  | NL-NL | 82.94 | 81.82 | 82.20 | 81.65 |
|  | FL-NL | 62.12 | 54.78 | 52.95 | 51.55 |
| GPT-oss (RL) | FL-FL | 79.28 | 75.51 | 73.95 | 76.97 |
|  | NL-NL | 75.66 | 75.99 | 75.77 | 77.14 |
|  | FL-NL | 53.83 | 49.03 | 47.42 | 49.08 |

*Table 4.* K&K CKA diagnostics. Cross-representation CKA decreases as logical complexity increases, while within-representation CKA remains stable under fixed templates.

| Model | Transfer Var. | Func. Top-10 FL/NL |
|---|---|---|
| Qwen3 (Orig.) | 15.68 | 4/5 |
| GPT-oss (Orig.) | 22.73 | 3/3 |
| Qwen3 (RL) | 11.48 | 4/4 |
| GPT-oss (RL) | 18.53 | 2/2 |

*Table 5.* K&K learning-transfer and attribution diagnostics.

## G.2. Reverse-Direction and Joint Training

The weak coupling of FL and NL learning is bidirectional. When training exclusively on NL and tracking FL accuracy, the reverse transfer ratio $\Delta$FL$/\Delta$NL also exhibits large fluctuations and frequent sign changes.

| Model | Ratio Range | Reverse Transfer Var. | FL-to-NL Var. |
|---|---|---|---|
| Qwen3-30B-A3B | $[-3.52, 6.32]$ | 9.37 | 15.68 |
| GPT-oss-20B | $[-3.52, 5.23]$ | 8.16 | 22.73 |

*Table 6.* Reverse-direction learning dynamics under NL-only training.

We also train jointly on FL and NL solving instances. Joint exposure improves NL accuracy, but it does not fuse representations: cross-representation CKA remains almost unchanged, transfer remains unstable, and attribution signatures persist.

| Diagnostic | Original | Joint-Trained | RL-Aligned |
|---|---|---|---|
| NL Perf. (Bool avg./CSP/Abel) | 28.2/36.4/62.6 | 67.4/78.8/79.6 | 75.2/80.2/83.2 |
| FL-NL CKA | 3.91 | 3.93 | 3.88 |
| Transfer Var. | 15.68 | 14.72 | 11.48 |
| Func. Top-10 (FL/NL) | 4/5 | 4/5 | 4/4 |

*Table 7.* Joint FL+NL training improves behavior but does not induce representational fusion.

