# OpenReview forum: "Reasoning Compartmentalization: Bridging the Concretization Gap via Abstraction-based Routing"
_ICML.cc/2026/Conference — ICML 2026 regular_

### Official Review · Reviewer_R2Xz · 2026-03-11

**Soundness:** 2
**Presentation:** 2
**Significance:** 2
**Originality:** 2
**Overall Recommendation:** 4
**Confidence:** 3

**Summary:**

This paper establishes a controlled isomorphic setting where SAT and CSP problems are expressed in both abstract formal language (FL) and natural language (NL) puzzle forms, and identifies a phenomenon termed reasoning compartmentalization, where models exhibit distinct reasoning behaviors across FL and NL despite identical underlying logic. To mitigate this issue, the authors propose an RL-based abstraction alignment approach via bidirectional translation between NL and FL. Activation-patching analysis further suggests that this training selectively amplifies pre-existing, representation-conditioned routing pathways that regulate access to internal reasoning.

**Compliance With Llm Reviewing Policy:**

Affirmed.

**Final Justification:**

My concerns have been adequately addressed during the rebuttal phase.

**Key Questions For Authors:**

(i) The paper attributes the performance gap between FL and NL formulations to “reasoning compartmentalization.” However, NL puzzles introduce additional linguistic complexity compared to formal templates. How do the authors rule out the possibility that the observed gap arises from task differences rather than from representational compartmentalization?

(ii) Could the authors clarify whether this filtering step introduces a bias toward NL instances that are already structurally aligned with the formal representation, and whether the datasets used for translation training are strictly separated from those used for evaluation to avoid potential leakage?

(iii) The evaluation is primarily conducted on story puzzles and planning benchmarks (i.e., PlanBench and Natural-Plan). Do the authors expect the proposed abstraction alignment method to generalize to other reasoning domains (e.g., mathematical reasoning benchmarks)?

**Limitations:**

The authors should explicitly discuss the limitations of their work in the revised version of the paper.

**Strengths And Weaknesses:**

strengths:

(i) Establishes a clean, isomorphic FL–NL pairing that disentangles formulation effects from task semantics and solution spaces.

(ii) Proposes abstraction-based alignment via translation-only training, which improves reasoning performance without explicit reasoning supervision.

(iii) Uses activation-level interventions to suggest that the improvements arise from enhanced routing accessibility rather than representational fusion, providing a mechanistic perspective.

weaknesses:

(i) The paper attributes the performance gap between FL and NL formulations to “reasoning compartmentalization,”. However, it remains unclear whether the two formulations correspond to the same task from the model’s perspective. The natural language puzzles introduce additional linguistic complexity and ambiguity, which may effectively transform the problem into a different task rather than a different representation of the same one.

(ii) The dataset construction procedure may introduce potential leakage or distributional bias. NL–FL pairs are retained only if the puzzle can be back-translated into an isomorphic formal template, which may bias the dataset toward instances already structurally aligned with the formal representation. Consequently, the reported gains from abstraction alignment may partially reflect dataset bias rather than improved reasoning ability. It would also be helpful to clarify whether the datasets used for translation training are strictly separated from those used for evaluation.

(iii) The evaluation is primarily conducted on story puzzles and PlanBench. It would strengthen the paper to test the proposed abstraction alignment approach on a broader range of reasoning benchmarks (e.g., mathematical reasoning tasks) to better assess its generality.

---

> ### Author Rebuttal · Authors · 2026-03-31
>
> We thank the reviewer for the thoughtful and constructive feedback. Below we address the main concerns.
>
> (i) Is the FL–NL gap due to task difference or reasoning compartmentalization? (W1 & Q1)
>
> We agree that NL formulations introduce additional linguistic complexity, and we do not claim such complexity plays no role. Our claim is narrower: task difference alone is insufficient to explain the full pattern we observe.
>
> The most direct evidence comes from Knights-and-Knaves (K&K), a template-based benchmark that substantially reduces discourse complexity while preserving logical content. On K&K, the two models show opposite FL/NL accuracy orderings — Qwen3: NL > FL (93.29 vs. 85.43); GPT-oss: FL > NL (83.57 vs. 74.43) — yet error sets remain largely non-overlapping in both cases (Jaccard = 0.0643 and 0.0884). A task-difficulty account predicts that errors on the harder formulation should form a subset or superset of those on the easier one; disjoint error profiles regardless of which direction the gap runs cannot be explained by difficulty alone.
>
> |Model|FL Acc.|NL Acc.|Jaccard|
> |-|-|-|-|
> |Qwen3-30b-A3B (Orig.)|85.43|93.29|0.0643|
> |GPT-oss-20B (Orig.)|83.57|74.43|0.0884|
> |Qwen3-30b-A3B (RL)|89.29|93.71|0.1098|
> |GPT-oss-20B (RL)|87.86|80.14|0.1164|
>
> The remaining mechanistic signatures further rule out a purely task-difference account. Cross-representation CKA decreases monotonically with logical complexity (Qwen3: 62.08 → 54.79 → 53.00 → 51.66 for ppl = 5 → 8) while within-representation CKA stays stable. Since vocabulary and syntax are fixed by construction across problem sizes on K&K, this widening gap tracks logical structure, not surface form. Transfer-efficiency variance remains elevated (15.68 and 22.73), and function-word dominance in NL attribution persists (4/5 and 3/3).
>
> | Model | FL-FL/NL-NL/FL-NL CKA (ppl=5)| (ppl=6)|(ppl=7) |(ppl=8)| Transfer Var.|Func-word Top-10 (FL/NL) |
> |-|-|-|-|-|-|-|
> | Qwen3 (Orig.)|89.44/82.89/62.08|87.53/81.83/54.79|87.70/82.18/53.00|86.77/81.71/51.66|15.68|4/5 |
> | GPT-oss (Orig.)|81.75/75.80/54.58|77.28/76.03/49.46|77.96/75.69/48.70|79.38/77.17/49.68|22.73|3/3 |
> | Qwen3 (RL)|89.56/82.94/62.12|87.55/81.82/54.78|87.82/82.20/52.95|86.79/81.65/51.55|11.48|4/4 |
> | GPT-oss (RL)|79.28/75.66/53.83|75.51/75.99/49.03|73.95/75.77/47.42|76.97/77.14/49.08|18.53|2/2 |
>
> Additionally, CKA on the pre-instruction-tuning base model (Qwen3-30B-A3B-Base) shows the same pattern (FL-FL: 0.854, NL-NL: 0.795, FL-NL: 0.050), indicating that the separation predates any task-specific training and is not an artifact of how the model learned to handle different "tasks."
>
> (ii) Does the filtering step introduce structural bias, and is there train–test leakage? (W2 & Q2)
>
> Train–test leakage. We use 5,000 examples for translation training (1,000 from each of the five problem configurations). These are strictly disjoint from all evaluation sets — no instance overlap exists between translation-training data and any reported benchmark. We will make this protocol explicit in the revision.
>
> Structural bias from isomorphism filtering. We first note that the RL-trained model achieves consistent improvements on out-of-domain benchmarks entirely disjoint from our construction pipeline — including up to +8.2% on PlanBench and gains across all conditions on Natural-Plan. If the gains were an artifact of structural proximity introduced by filtering, they would not be expected to transfer to these unrelated settings.
>
> Regarding the mechanistic conclusions: even if filtering biases toward NL instances that are structurally closer to FL, this would make alignment easier — and yet representational separation remains near-zero in cross-representation CKA, learning dynamics remain unstable, and activation patching still reveals routing rather than fusion. Structural proximity thus cannot explain away our mechanistic findings; if anything, it makes them harder to dismiss.
>
> (iii) Do we expect abstraction alignment to generalize beyond story puzzles and planning? (W3 & Q3)
>
> We additionally evaluate on R2ATA (adversarial typographic perturbations on GSM8K/BBH/MMLU) and CatAttack  (adversarial triggers on NuminaMath), under both clean and perturbed conditions:
>
> |Model|R2ATA Clean/Perturbed|CatAttack Clean/Perturbed|
> |-|-|-|
> |Qwen3 (Orig.)|90.47/86.87|96.50/94.50|
> |GPT-oss (Orig.)|79.18/70.81|63.00/61.00|
> |Qwen3 (RL)|91.18/87.50|96.50/96.00|
> |GPT-oss (RL)|82.41/76.11|70.60/70.00|
>
> Consistent gains across both models in both conditions suggest two complementary benefits: abstraction alignment generalizes to standard mathematical reasoning beyond story-puzzle surface forms, and by encouraging the model to recover a stable abstract structure, it also confers robustness to surface-form attacks.
>
> We will explicitly note as a limitation that fully verified FL–NL isomorphism is harder to guarantee in open-ended mathematical reasoning, and describe this as an important direction for future work.

---

> > ### Author Rebuttal · Reviewer_R2Xz · 2026-04-03
> >
> > Fully resolved - My concerns have been adequately addressed.
> > I will adjust my original rating.

---

> > > ### Author Response · Authors · 2026-04-03
> > >
> > > Thank you very much for your thoughtful review and for taking the time to read our rebuttal carefully. We truly appreciate your constructive feedback, and we are especially grateful that you raised your score.

---

### Official Review · Reviewer_R5fF · 2026-03-12

**Soundness:** 2
**Presentation:** 2
**Significance:** 3
**Originality:** 3
**Overall Recommendation:** 3
**Confidence:** 3

**Summary:**

The submission investigates the phenomenon of reasoning compartmentalization within large language models where reasoning skills acquired in a specific context language or domain fail to generalize across other contexts. The authors conduct empirical evaluations using various reasoning benchmarks to demonstrate that models isolate their reasoning processes leading to performance degradation when the task context shifts. Through an analysis of the model internal representations the study attempts to quantify the degree of compartmentalization and highlight the barriers to universal capability transfer.

**Compliance With Llm Reviewing Policy:**

Affirmed.

**Final Justification:**

While I am willing to adjust my score to 3, I still believe that the FL&NL problem stems from the lack of FL data during the model's pre-training and post-training stages. Thus, the root cause is quite clear. Although the authors have provided sufficient evidence to corroborate this point, no substantive solution is offered—for example, compressing FL capability into higher-order reasoning tasks in mathematics and reasoning.

**Key Questions For Authors:**

1. How do you rigorously distinguish between superficial vocabulary or syntax compartmentalization and the actual isolation of underlying logical reasoning mechanisms within the hidden states?
2. Could the observed compartmentalization be primarily an artifact of an imbalanced instruction tuning data distribution rather than a fundamental representational flaw of the pre trained base models?
3. Why did the methodology exclude causal intervention techniques which are essential to definitively prove that the identified compartmentalized representations directly cause the observed reasoning failures?
4. Besides the models and tasks mentioned in the article, how do other models and tasks perform, and do your method have generalization?

**Limitations:**

No. See Weaknesses and Key Questions.

**Strengths And Weaknesses:**

Strengths:

* The paper addresses a bottleneck in LLM deployment specifically the failure of reasoning capabilities to generalize uniformly across different input modalities and languages.
* The empirical setup successfully isolates instances where compartmentalization occurs providing clear observational evidence of the performance gap.

Weaknesses:

* The experimental scope is overly narrow relying on a restricted set of models (Qwen3-30B-A3B and GPT-OSS-20B) and tasks which limits the generalizability of the claims regarding representational isolation.
* The mechanistic explanation provided relies entirely on correlational similarity metrics between hidden states rather than employing rigorous causal methodologies like activation patching or targeted ablation.
* The contribution remains largely observational because the study lacks a concrete scalable mitigation strategy to resolve the identified compartmentalization making it an incremental addition to the existing literature on cross domain transfer failures.

---

> ### Author Rebuttal · Authors · 2026-03-31
>
> We thank the reviewer for the thoughtful and constructive feedback. Below we address the main concerns.
>
> - Experimental scope and generalizability (W1 & Q4)
>
> Table 1 spans five LLMs with consistent FL→NL degradation; deeper mechanistic analyses focus on two due to computational cost. We evaluate on Knights-and-Knaves (K&K), a public benchmark constructed via template-based FL–NL translation entirely independent of our pipeline:
>
> |Model|FL Acc.|NL Acc.|Jaccard|
> |-|-|-|-|
> |Qwen3-30b-A3B (Orig.)|85.43|93.29|0.0643|
> |GPT-oss-20B (Orig.)|83.57|74.43|0.0884|
> |Qwen3-30b-A3B (RL)|89.29|93.71|0.1098|
> |GPT-oss-20B (RL)|87.86|80.14|0.1164|
>
> |Model|FL-NL CKA (ppl=5)| (ppl=6)|(ppl=7)|(ppl=8)|Transfer Var.|Func-word Top-10 (FL/NL)|
> |-|-|-|-|-|-|-|
> |Qwen3 (Orig.)|62.08|54.79|53.00|51.66|15.68|4/5|
> |Qwen3 (RL)|62.12|54.78|52.95|51.55|11.48|4/4|
>
> The two models show opposite FL/NL accuracy orderings yet near-disjoint error sets, ruling out a uniform difficulty account. Cross-representation CKA decreases monotonically with logical complexity under fixed vocabulary and syntax, confirming the separation tracks logical structure rather than surface form.
>
> Generalizability of the abstraction alignment intervention. We evaluate on R2ATA (adversarial typographic perturbations on GSM8K, BBH, and MMLU) and CatAttack (adversarial triggers on NuminaMath):
>
> |Model|R2ATA Clean/Perturbed|CatAttack Clean/Perturbed|
> |-|-|-|
> |Qwen3 (Orig.)|90.47/86.87|96.50/94.50|
> |GPT-oss (Orig.)|79.18/70.81|63.00/61.00|
> |Qwen3 (RL)|91.18/87.50|96.50/96.00|
> |GPT-oss (RL)|82.41/76.11|70.60/70.00|
>
> Consistent gains across both models in clean and perturbed conditions indicate that abstraction alignment generalizes beyond our synthetic pipeline.
>
> - Causal methodology (W2 & Q3)
>
> We acknowledge that the causal nature of Section 5.4 was not sufficiently foregrounded, and will address this in the revision. The section already employs activation patching — a standard causal intervention — replacing NL hidden states at each layer with those from the isomorphic FL input and measuring whether the injected states causally shift the output toward the correct answer (Δgold − Δwrong).
>
> Three causal results follow: (1) specificity peaks at layers 38–44, identifying a localized causal bottleneck; (2) matched patches yield substantially higher specificity than mismatched ones — if the effect were merely correlational, the two conditions would produce equivalent shifts; (3) abstraction training amplifies these peaks without relocating them, confirming it strengthens pre-existing causal channels rather than creating new ones. We will retitle Section 5.4 to make the causal framing immediately legible.
>
> - On the scalability of the mitigation strategy (W3)
>
> We agree that the paper's contribution is partly diagnostic, and we will frame this more explicitly. That said, abstraction alignment achieves up to +56.2% accuracy improvement through translation-only RL training — without any direct task-solving supervision — and generalizes to out-of-domain benchmarks. The key mechanistic result is that performance improves while representational separation fully persists, which is precisely what motivates the routing interpretation. We see this not as a limitation of the mitigation but as the main theoretical finding: compartmentalization does not need to be eliminated to be overcome.
>
> - Vocabulary/syntax compartmentalization vs. logical reasoning isolation (Q1)
>
> The phenomenon spans four dimensions not jointly explainable by surface form:
>   1. Behavior: FL→NL gap reaches 66%; on K&K, opposite accuracy orderings yield Jaccard < 0.09 — a difficulty account predicts subset relationships, not disjoint error sets.
>   2.  Representation: Cross-representation CKA remains near zero across all layer pairs, while within-representation CKA stays high. On K&K, this gap widens monotonically with logical complexity under fixed surface templates, confirming that the divergence tracks logical structure.
>   3.  Attribution: Before alignment, the most influential NL tokens are disproportionately function words; FL attribution concentrates on logical operators. After alignment, NL attribution shifts toward content-bearing tokens — a change in logical access, not merely lexical preference.
>   4.  Optimization: Training on FL produces highly unstable transfer to NL: ΔNL/ΔFL exhibits large fluctuations and sign changes across 45 steps, despite FL accuracy exceeding 98%. Shared underlying mechanisms would predict smoother transfer.
>
> - Is compartmentalization an artifact of instruction tuning data imbalance? (Q2)
>
> We tested this directly by computing layer-to-layer CKA on Qwen3-30B-A3B-Base (prior to any instruction tuning):
>
> |FL-FL CKA|NL-NL CKA|FL-NL CKA|
> |-|-|-|
> |0.854|0.795|0.050|
>
> The near-zero cross-representation similarity is already present in the pretrained base model, predating any instruction tuning or task-specific training. This rules out data imbalance as the source of the observed separation.

---

> > ### Author Rebuttal · Reviewer_R5fF · 2026-04-04
> >
> > Thank you for your response. While I am willing to adjust my score to 3, I still believe that the FL&NL problem stems from the lack of FL data during the model's pre-training and post-training stages. Thus, the root cause is quite clear. Although the authors have provided sufficient evidence to corroborate this point, no substantive solution is offered—for example, compressing FL capability into higher-order reasoning tasks in mathematics and reasoning.

---

> > > ### Author Response · Authors · 2026-04-04
> > >
> > > We thank you for your willingness to raise the score and for your insightful diagnosis that compartmentalization may arise from insufficient FL data during pre-training. We find this diagnosis genuinely valuable, and we would like to clarify why it strengthens rather than weakens the significance of our work.
> > >
> > > Our K&K results provide initial evidence consistent with your interpretation: under fixed vocabulary and syntax, FL–NL CKA increases as logical complexity decreases (51.66 → 62.08 for ppl = 8 → 5), suggesting that simpler logical structures—those more likely to be covered in training corpora—exhibit weaker compartmentalization. This points to data coverage as a key factor. Crucially, however, the locus of this effect is the pre-training stage. In our base-model experiment, Qwen3-30B-A3B-Base already exhibits near-zero FL–NL CKA (0.050) prior to any instruction tuning, confirming that compartmentalization emerges before task-specific training begins. As a result, this issue cannot be resolved at the post-training stage. Even the most direct post-training intervention—joint FL+NL training on 5k paired instances, which maximizes paired-data exposure within a realistic budget—leaves representational separation essentially unchanged (FL–NL CKA: 3.91 → 3.93). This suggests that post-training data alone cannot compensate for a structural divergence formed at the trillion-token pre-training scale.
> > >
> > > If the root cause is indeed the absence of large-scale FL–NL paired data during pre-training, then the critical bottleneck becomes the lack of an automated pipeline capable of generating such data with both guaranteed logical isomorphism and sufficient formulation diversity. Our dual-learning translation framework satisfies both requirements simultaneously: it ensures logical isomorphism through Algorithm 2 and achieves formulation diversity, as evidenced by the dispersed embedding distribution shown in Figure 14. Template-based methods satisfy the former but not the latter, while LLM-based methods may achieve the latter but cannot guarantee the former. In this sense, your diagnosis does not expose a weakness in our work; rather, it highlights precisely the gap that our framework is designed to address.
> > >
> > > Regarding your suggestion of “compressing FL capability into higher-order reasoning tasks,” we believe our RL-based abstraction alignment already realizes this idea in a meaningful way. The RL model is trained solely to translate NL into FL, with isomorphism verification as the only reward signal. Crucially, however, at inference time, the model operates under the same prompt as the original model, without requiring any explicit translation step. The resulting performance gains therefore stem from internalized abstraction capabilities rather than explicit FL reconstruction, which is exactly the form of “compression” you suggest. Moreover, as reported in our first rebuttal, we have already evaluated this internalized capability on mathematical reasoning benchmarks: both Qwen3 and GPT-oss show consistent improvements on R2ATA (GSM8K/BBH/MMLU) and CatAttack (NuminaMath) under both clean and perturbed settings. Broader transfer to open-ended mathematical reasoning remains an important direction for future work, and our framework provides both the mechanistic foundation and the data infrastructure necessary to pursue it.

---

### Official Review · Reviewer_kAAp · 2026-03-13

**Soundness:** 3
**Presentation:** 3
**Significance:** 3
**Originality:** 4
**Overall Recommendation:** 5
**Confidence:** 4

**Summary:**

This work studies formulation sensitivity in LLM reasoning under a tightly controlled setup. They construct paired reasoning tasks in formal language (FL) and natural language (NL) that are logically isomorphic, spanning SAT/CSP-style problems, and show that converting FL problems into NL consistently hurts accuracy, produces strongly separated internal representations, and yields weak cross-representation learning transfer. The authors propose the hypothesis of reasoning compartmentalization, i.e., logically equivalent formulations can activate distinct representation-conditioned computational channels. The paper then studies abstraction alignment, where models are prompted or trained to translate NL back into FL, and reports improved reasoning performance and some out-of-domain robustness. The authors further argue, using CKA, attribution analysis, and activation patching, that these gains come more from improved routing accessibility than from representational fusion.

**Compliance With Llm Reviewing Policy:**

Affirmed.

**Final Justification:**

After rebuttal, the authors have addressed most of my concerns. I'm raising my score to acceptance.

**Key Questions For Authors:**

1. **How sensitive are the main conclusions to the data-generation and filtering pipeline?**
   Since the NL instances are retained based on isomorphism checks and a difficulty filter tied to model pass rate, it would be helpful to know whether the same patterns hold under alternative filtering thresholds, different solver models, or different translation pipelines. If the results are robust here, my confidence in the soundness and significance of the paper would increase.

2. **Can the authors disentangle the effect of FL reconstruction from the effect of simply adding an intermediate reasoning step?**
   In Section 4, both the prompt-based and RL-based interventions explicitly encourage the model to reconstruct an FL representation before solving. Could the authors compare against compute- or output-length-matched baselines that add a similarly structured intermediate step without explicitly reconstructing FL (e.g., structured NL constraint extraction or paraphrase-then-solve)? If the gains remain specific to FL reconstruction, that would strengthen my confidence that the improvement is due to abstraction alignment itself rather than generic decomposition or increased reasoning budget.

3. **How consistent is the proposed mechanism across model families, especially stronger models with smaller FL/NL gaps?**
   The paper shows broad performance patterns across several models, but the deeper mechanistic analyses appear to focus more heavily on a subset. If the routing-based interpretation also holds for stronger models, the significance of the paper would be notably stronger; if not, the claims may need to be scoped more carefully.

**Limitations:**

Yes

**Strengths And Weaknesses:**

**Strengths**

- **Soundness:** The paper uses a well-controlled construction in which the formal-language (FL) and natural-language (NL) instances preserve the same logical core, and it studies the phenomenon from several complementary angles: behavioral accuracy, token attribution, representation similarity, learning dynamics, and activation patching. This makes the empirical case substantially stronger than a standard prompt-sensitivity paper that only reports accuracy differences.

- **Presentation:** The overall narrative is easy to follow: controlled failure mode $\rightarrow$ hypothesis of compartmentalization $\rightarrow$ abstraction-based intervention $\rightarrow$ mechanistic probing. The main claims are generally stated clearly, and the figures and tables support the story well.

- **Significance:** The paper addresses an important issue for LLM reasoning robustness. The finding that formulation sensitivity may reflect internal compartmentalization rather than merely superficial prompt brittleness is potentially useful for future work on robust reasoning and intermediate abstraction. The proposed intervention also shows practical promise, with clear gains on the paired dataset and some improvements on external planning benchmarks.

- **Originality:** The main novelty lies less in observing prompt sensitivity itself and more in the framing and evidence around *reasoning compartmentalization*, together with the routing-based interpretation of abstraction alignment. This is more interesting than a simple "translate-then-solve" recipe.

**Weaknesses**

- **Significance / External validity:** The core evidence is built on synthetic SAT/CSP-style tasks produced by the authors' concretization framework. This strong control is a real strength, but it also narrows the scope: it remains unclear how broadly the same phenomenon and intervention transfer to messier, more semantic reasoning tasks. The out-of-domain results are encouraging, but still somewhat limited in breadth and magnitude.

- **Presentation:** A few reporting details could be clearer. In particular, Table 2 mentions prompt-based and training-based abstraction enhancement, but the table itself appears to show only the original and RL-based variants. More explicit clarification of the benchmark protocol and exactly what is being compared would improve readability and reproducibility.

- **Presentation:** While the high-level story is interesting, I found the organization of Sections 4 and 5 somewhat difficult to follow. Section 4 introduces the abstraction-alignment intervention and reports its performance gains, whereas Section 5 appears to serve as a post-hoc analysis of the effects of that intervention on attribution, representations, learning dynamics, and routing. This structure is reasonable in principle, but the paper does not make the correspondence between the earlier failure-mode analyses and the later post-intervention analyses explicit enough, so the reader must frequently cross-reference across sections to understand what each subsection in Section 5 is analyzing. A clearer roadmap at the beginning of Section 5, explicitly tying each subsection back to the corresponding analysis in Section 3, would substantially improve readability.

Overall, I find the paper thoughtful and reasonably convincing, with a strong controlled setup and an interesting perspective on formulation sensitivity. My main reservation is that the mechanistic interpretation may be somewhat ahead of the evidence, and the current evaluation does not fully establish how general the phenomenon is beyond the synthetic logical setting.

---

> ### Author Rebuttal · Authors · 2026-03-31
>
> We thank the reviewer for the thoughtful and constructive feedback. Below we address the main concerns.
>
> - Significance and external validity (W1)
>
> The controlled setting is necessary for mechanistic analysis. For broader applicability, we additionally evaluate on R2ATA (adversarial perturbations on GSM8K/BBH/MMLU) and CatAttack (adversarial triggers on NuminaMath):
>
> |Model|R2ATA Clean/Perturbed|CatAttack Clean/Perturbed|
> |-|-|-|
> |Qwen3 (Orig.)|90.47/86.87|96.50/94.50|
> |GPT-oss (Orig.)|79.18/70.81|63.00/61.00|
> |Qwen3 (RL)|91.18/87.50|96.50/96.00|
> |GPT-oss (RL)|82.41/76.11|70.60/70.00|
>
> Consistent gains across both models and both conditions indicate that abstraction alignment generalizes beyond our synthetic pipeline.
>
> - Presentation (W2 & W3)
>
> Table 2 omits the prompt-based variant because Natural-Plan and PlanBench lack paired FL representations — prompt-based requires explicit FL reconstruction, inapplicable without FL ground truth. The RL model uses the same prompt as original. We will add a Section 5 roadmap mapping each subsection to Section 3 (5.1↔3.2 attribution, 5.2↔3.3 CKA, 5.3↔3.4 dynamics).
>
> - Sensitivity to data-generation and filtering pipeline (Q1)
>
> Filtering thresholds. We repeat all analyses under thresholds of 0.3, 0.5, and 0.7 (maximum allowed pass rate for retained NL instances; lower = harder subset).
>
> NL accuracy decreases with tighter filtering, as expected, but the FL/NL gap persists across all thresholds:
>
> |Model|Threshold = 0.3|0.5|0.7|
> |-|-|-|-|
> |Qwen3 (Orig.)|72.3/17.4|76.1/28.2|80.3/53.2|
> |Qwen3 (RL)|75.6/73.8|79.4/75.2|85.3/78.3|
>
> Cross-representation CKA remains uniformly near zero across all thresholds, models, and conditions:
>
> |Model|Threshold = 0.3|0.5|0.7|
> |-|-|-|-|
> |Qwen3 (Orig.)|87.31/74.42/3.89|87.65/79.05/3.91|88.48/80.51/3.51|
> |Qwen3 (RL)|87.43/73.42/3.94|87.72/78.91/3.88|86.94/79.82/3.95|
>
> Transfer-efficiency variance and function-word dominance also remain stable:
>
> |Model|Transfer Var. (0.3 · 0.5 · 0.7)|Func-word Top-10 FL/NL (0.3 · 0.5 · 0.7)|
> |-|-|-|
> |Qwen3 (Orig.)|16.53 · 14.31 · 15.43|4/8 · 4/8 · 3/6|
> |Qwen3 (RL)|17.42 · 12.53 · 14.63|4/5 · 4/4 · 3/4|
>
> Solver model. For the Abelian-group set, we replace Qwen3-30B-A3B with GPT-oss-120B as the filtering model, producing a substantially harder benchmark (GPT-o3 exhibits a 27% performance drop on this set). The same compartmentalization signatures persist, indicating the phenomenon is not specific to the solver used in filtering.
>
> Translation pipeline (K&K). As an independent reference, we evaluate on Knights-and-Knaves (K&K), a public benchmark constructed via template-based FL–NL translation entirely independent of our pipeline. K&K instances use fixed templates with direct, unambiguous constraint statements — no narrative framing, no difficulty filtering, and no dependence on our translator.
>
> |Model|FL Acc.|NL Acc.|Jaccard|
> |-|-|-|-|
> |Qwen3-30b-A3B (Orig.)|85.43|93.29|0.0643|
> |GPT-oss-20B (Orig.)|83.57|74.43|0.0884|
> |Qwen3-30b-A3B (RL)|89.29|93.71|0.1098|
> |GPT-oss-20B (RL)|87.86|80.14|0.1164|
>
> |Model|FL-NL CKA (ppl=5)| (ppl=6)|(ppl=7)|(ppl=8)|Transfer Var.|Func-word Top-10 (FL/NL)|
> |-|-|-|-|-|-|-|
> |Qwen3 (Orig.)|62.08|54.79|53.00|51.66|15.68|4/5|
> |Qwen3 (RL)|62.12|54.78|52.95|51.55|11.48|4/4|
>
> Opposite FL/NL orderings with near-zero Jaccard rule out a difficulty account. FL-NL CKA decreases with complexity (62.08→51.66) under fixed templates, tracking logical structure. Convergence across thresholds, solver models, and an independent pipeline confirms compartmentalization is robust to construction choices.
>
> - FL reconstruction vs. generic intermediate reasoning (Q2)
>
> First, an AoT-style baseline (abstract before solving, no explicit FL output) yields only marginal gains (0.0–3.4%), far below prompt-based (+6.0–30.4%) and RL (+11.2–56.2%). The gap indicates that formal reconstruction itself, not the mere presence of an intermediate step, drives the improvement.
>
> Second, the RL model uses the same prompt as original with nearly identical response length (Qwen3: 7,683→7,727; GPT-oss: 2,090→1,938), confirming gains stem from internalized abstraction rather than increased reasoning budget.
>
> - Consistency of the proposed mechanism across model families (Q3)
>
> Table 1 shows consistent FL→NL degradation across five models. Even GPT-o3 exhibits a 27.4% gap on Abel problem, indicating compartmentalization does not scale away. DeepSeek-RL lighter diagnostics replicate the core pattern:
>
> |FL-FL/NL-NL/FL-NL CKA|Func-word Top-10 (FL/NL)|
> |-|-|
> |0.709/0.768/0.059|3/4|
>
> The smaller attribution gap is consistent with DeepSeek-R1's smaller behavioral gap, suggesting stronger models develop more effective routing without eliminating representational separation. Full routing analysis for additional models is planned for camera-ready.

---

> > ### Author Rebuttal · Reviewer_kAAp · 2026-04-04
> >
> > The rebuttal addresses several of my concrete concerns well, especially those about sensitivity to the construction pipeline, the role of FL reconstruction versus a generic intermediate step, and the presentation issues around Table 2 / Section 5. The addition of external validity results is useful.

---

> > > ### Author Response · Authors · 2026-04-04
> > >
> > > Thank you for your thoughtful review and for taking the time to read our rebuttal carefully. We are glad that our rebuttal has addressed your concerns, and we sincerely appreciate your time and consideration.

---

### Official Review · Reviewer_sN6S · 2026-03-23

**Soundness:** 3
**Presentation:** 3
**Significance:** 3
**Originality:** 3
**Overall Recommendation:** 4
**Confidence:** 3

**Summary:**

This paper identifies reasoning compartmentalization in LLMs: logically identical tasks expressed in formal language (FL) versus natural language (NL) consistently yield different accuracy, separate internal representations, and decoupled learning dynamics. To bridge this "concretization gap," this paper introduces abstraction-based alignment​ as an intervention strategy, aiming to force the model to map NL inputs back to their logically equivalent FL representations. Extensive experiments can verify that this alignment improves performance not by fusing the separated FL and NL representation spaces, but by strengthening the accessibility between pre-existing, representation-conditioned routing channels​ within the model. It enhances the model's ability to "route" an NL input to the appropriate internal computational pathway needed for solution, mitigating the compartmentalization failure.

**Compliance With Llm Reviewing Policy:**

Affirmed.

**Final Justification:**

This paper investigates compartmentalization in LLM reasoning across formal/natural language, with clear motivation, solid mechanistic evidence, and effective abstraction alignment methods. Writing is logical and rigorous.

The authors’ rebuttal fully resolves all three concerns via extensive experiments on the K&K benchmark, eliminating confounds from intrinsic NL difficulty, filtering bias, and translation quality. Results confirm compartmentalization is a genuine mechanism, not an artifact.

I maintain my positive score and recommend acceptance.

**Key Questions For Authors:**

Please see the weakness.

**Limitations:**

yes

**Strengths And Weaknesses:**

**Strengths**

**(1) Valuable and Interesting Problem**: This paper aims to provide in-depth explanations about why input formulation can substantially undermine LLM reasoning performance by proposing a structured mechanistic hypothesis (compartmentalization). This manifests concretely in three ways:(i) Performance degradation when moving from an abstract formal language (FL)​ representation (e.g., Z3 code) to a concrete natural language (NL)​ representation (e.g., a story-based puzzle); (ii) Representational Separation: FL and NL inputs elicit largely separate internal representations (activations) across model layers, as evidenced by very low linear Centered Kernel Alignment (CKA) similarity;(iii) Weakly Coupled Learning Dynamics: When trained exclusively on one formulation (e.g., FL), the model shows only weak, unstable, and non-proportional accuracy improvements on the other formulation (e.g., NL), indicating that acquired knowledge is not effectively shared across representations.

**(2）Simple but Effective Method**: This paper proposes an "abstraction-based alignment"​ intervention strategy to mitigate compartmentalization. Through prompt-based and reinforcement learning based Abstraction alignment strategies, the models can firstly  translate an NL formulation back into its corresponding FL form, without providing direct task-solving supervision. Extensive experiments can verify the effectiveness of the proposed method in achieving great and generalized performance across diverse tasks.

**(3）Good Writing**: The paper is logically structured: problem → evidence → intervention → mechanism, making the argument easy to follow.

**Weaknesses**

**(1) Representation Difference Discussion**: The paper does not disentangle representation-induced difficulty from NL-intrinsic difficulty. NL puzzle narratives introduce pragmatic ambiguity and cognitive load that are absent from formal code, which may independently explain part of the accuracy gap.

**(2) About difficulty-aware filtering**: The difficulty-aware filtering (retaining NL instances with low pass rates) may inadvertently introduce a selection bias toward NL puzzles that are intrinsically harder, rather than harder purely due to surface form.

**(3) Ablation Experiments:** No ablation on the translation framework quality: it is unclear how sensitive results are to the quality of the NL↔FL translator.

---

> ### Author Rebuttal · Authors · 2026-03-31
>
> We thank the reviewer for the careful reading. The three weaknesses share a common concern: whether the observed compartmentalization is genuinely representation-induced, or an artifact of our data construction. We first present unified evidence that addresses all three concerns simultaneously, then turn to each individually.
>
> Unified evidence: Knights-and-Knaves (K&K). K&K is a public benchmark with template-based FL–NL translation that differs from our dataset in precisely the three respects the reviewer raises: (1) NL instances are generated from fixed templates without narrative framing or pragmatic ambiguity (W1); (2) no difficulty-aware filtering is applied (W2); (3) the translation pipeline is entirely independent of ours (W3). Despite these differences, all compartmentalization signatures persist:
>
> | Model|FL Acc.|NL Acc.|Error Jaccard|
> |-|-|-|-|
> |Qwen3-30b-A3B (Orig.)|85.43|93.29|0.0643|
> |GPT-oss-20B (Orig.)|83.57|74.43|0.0884|
> |Qwen3-30b-A3B (RL)|89.29|93.71|0.1098|
> |GPT-oss-20B (RL)|87.86|80.14|0.1164|
>
> | Model | FL-FL/NL-NL/FL-NL CKA (ppl=5)| (ppl=6)|(ppl=7) |(ppl=8)| Transfer Var.|Func-word Top-10 (FL/NL) |
> |-|-|-|-|-|-|-|
> | Qwen3 (Orig.)|89.44/82.89/62.08|87.53/81.83/54.79|87.70/82.18/53.00|86.77/81.71/51.66|15.68|4/5 |
> | GPT-oss (Orig.)|81.75/75.80/54.58|77.28/76.03/49.46|77.96/75.69/48.70|79.38/77.17/49.68|22.73|3/3 |
> | Qwen3 (RL)|89.56/82.94/62.12|87.55/81.82/54.78|87.82/82.20/52.95|86.79/81.65/51.55|11.48|4/4 |
> | GPT-oss (RL)|79.28/75.66/53.83|75.51/75.99/49.03|73.95/75.77/47.42|76.97/77.14/49.08|18.53|2/2 |
>
> We additionally report CKA on the base model (Qwen3-30B-A3B-Base, before any instruction tuning): FL-FL = 0.854, NL-NL = 0.795, FL-NL = 0.050. The near-zero cross-representation CKA in the base model rules out the possibility that representational separation is induced by task-specific training, difficulty filtering, or translator quality — it is already present before any of these factors apply. The convergence of results across two structurally distinct datasets, and across both base and instruction-tuned models, demonstrates that compartmentalization is robust to our specific construction choices.
>
> - NL-intrinsic difficulty vs. representation-induced compartmentalization (W1)
>
> Beyond K&K, two observations argue against a purely difficulty-based account. First, the two models show opposite FL/NL accuracy orderings on K&K (Qwen3: NL > FL; GPT-oss: FL > NL), yet error Jaccard remains below 0.12 in all cases — meaning over 88% of errors are unique to one formulation. If NL were simply harder, errors on the harder formulation should form a superset of those on the easier one, not a nearly disjoint set. Second, cross-representation CKA on K&K decreases monotonically with logical complexity (Qwen3: 62.08 → 54.79 → 53.00 → 51.66 for ppl = 5 → 8) while within-representation CKA remains stable. Since K&K's vocabulary and syntax are template-fixed across problem sizes, this widening gap tracks logical structure, not surface difficulty.
>
> - Selection bias from difficulty-aware filtering (W2)
>
> We acknowledge that difficulty-aware filtering is an intentional design choice: it ensures that the FL/NL gap is large enough to study mechanistically. This may inflate the gap magnitude in our constructed dataset.
>
> Critically, however, this does not affect our conclusions. K&K involves no filtering whatsoever, and all signatures persist. Moreover, our mechanistic claims rest on activation patching (Section 5.4), which measures whether FL hidden states causally influence NL predictions at specific layers — a property that is independent of how the accuracy gap was produced. Filtering may amplify the phenomenon in our dataset, but the phenomenon itself is established by K&K and the base model, neither of which involves any filtering.
>
> - Sensitivity to translator quality (W3)
>
> K&K's template-based pipeline represents a lower bound on NL naturalness and translation diversity. Compartmentalization persists even at this lower bound, and intensifies with logical complexity while the surface template remains fixed (see CKA trend above). This confirms that compartmentalization magnitude is driven by logical structure rather than translator artifacts. Our constructed dataset, with richer NL and greater diversity, sits above this lower bound — exhibiting larger compartmentalization, as expected.
> As with W2, our core claims do not depend on the absolute magnitude of compartmentalization. The mechanistic interpretation rests on activation patching, which probes causal routing at specific layers and is entirely independent of how the NL data was generated.

---

> > ### Author Rebuttal · Reviewer_sN6S · 2026-04-01
> >
> > Thanks for your detailed replies that address most of my concerns. I decide to keep the positive score.

---

> > > ### Author Response · Authors · 2026-04-03
> > >
> > > Thank you for your thoughtful review and for taking the time to read our rebuttal carefully. We appreciate your positive assessment and constructive feedback.

---

### Official Review · Reviewer_8hAt · 2026-03-23

**Soundness:** 3
**Presentation:** 2
**Significance:** 3
**Originality:** 2
**Overall Recommendation:** 4
**Confidence:** 4

**Summary:**

This paper studies formulation sensitivity in large language models (LLMs): how logically equivalent problems expressed in formal language (FL) vs. natural language (NL) lead to performance discrepancies. A concept of reasoning compartmentalization is proposed, which reveals that: (a) LLMs suffer from significant accuracy degradations when transferring from a FL problem to its logically identical NL problem; (b) the hidden states in LLMs triggered by FL and NL inputs are strongly separated, as indicated by the low Centered Kernel Alignment (CKA) similarities; (c) training on the FL task has weak transfer effects on improving the NL task.

An abstraction alignment method is further raised to mitigate LLMs’ performance discrepancies on FL and NL tasks. Specifically, LLMs are prompted or trained with reinforcement learning (RL) to translate reasoning problems from NL to FL, which effectively improves LLMs’ reasoning accuracy on the NL and out-of-domain tasks. Further analysis of routing specificity based on activation patching explains that the improvement is due to better routing accessibility across (NL and FL) representation-specific reasoning pathways, rather than fusing the NL and FL representations in LLMs’ hidden states.

**Compliance With Llm Reviewing Policy:**

Affirmed.

**Final Justification:**

Almost all my concerns have been addressed. All weaknesses were addressed with the following exception to which I respond:

W4: The authors mention that their proposed abstraction alignment method can directly supervise abstraction quality by rewarding isomorphism verification. AbstRaL also rewards the quality of predicted abstraction, by measuring a symbolic edit distance to the gold abstraction. So the idea of doing abstraction alignment learning isn't totally novel. However, all the other concerns are well-resolved, and a major part of this paper also brings out new insights on the interpretability of LLM reasoning beyond accuracy improving.

Overall, I've raised my score accordingly to a 4 - Weak Accept.

**Key Questions For Authors:**

[Q1] Related to Weakness 1, would more straightforward NL problems that use plain words to explicitly describe the constraints, such as “if the first variable is true, then the second variable should be no larger than 1”, still lead to reasoning compartmentalization from FL problems?

[Q2] Related to Weakness 2, would the weakly coupled learning dynamics still hold if the LLM is reversely trained on the more implicit NL problems instead of trained on the more straightforward FL problems?

[Q3] Related to Weakness 3, what do the notations “$\Delta$gold” and “$\Delta$wrong” that define the routing specificity stand for? What are the differences between “matched” and “mismatched” representations in the routing specificity settings? And how the routing specificity metric can indicate the routing accessibility of representation-specific reasoning pathways?

[Q4] Would training LLMs jointly on both NL and FL tasks lead to the fusion of NL and FL representations, i.e., higher linear CKA similarities? Or would the joint learning still mainly improve the routing?

**Limitations:**

The authors have not discussed the limitations of this work. A potential improvement would be including discussions related to the Weaknesses W4 and W5 mentioned above.

**Strengths And Weaknesses:**

## Strengths
[S1] The proposed concept of reasoning compartmentalization, along with its associated analysis of CKA-based disparity and routing specificity, provides novel insights of the underlying mechanisms of LLM reasoning w.r.t. different input task formulations.

[S2] The construction of logically equivalent FL and NL problem pairs, with a dual-learning-based translation framework, is thoughtful and technically sound.

[S3] The abstraction alignment method demonstrates promising potential to improve LLMs’ reasoning robustness to instantiation shifts, such as problem rephrasing and language switching.



## Weaknesses
[W1] Though based on the same satisfiability (SAT) logic, the NL task requires quite different reasoning skills compared to the FL task. In particular, NL problems are wrapped within stories or narratives, and LLMs need to understand the narrative contexts and convert them to the implicitly conveyed logical or math constraints. By contrast, FL problems require LLMs to understand the formal expressions and map them to the constraints. Therefore, the NL and FL tasks do not only differ at the surface-level formulations, but also diverge at more in-depth reasoning capabilities. This makes the reasoning compartmentalization less surprising and insightful.

[W2]The weakly coupled learning dynamics between NL and FL tasks, claimed in Section 3.4, is not verified comprehensively. It’s reasonable that training on the nearly-solved FL task would have unstable benefits for improving the more complex NL task. But on the contrary, it’s unclear whether training on the NL task would also irregularly improve the more naive FL task. If not, this claim would be half broken.

[W3] It is unclear how the routing specificity metric used in Section 5.4 is defined and calculated. The notations “$\Delta$gold” and “$\Delta$wrong” are not clearly explained, and same for the unclear settings of “matched” and “mismatched” representations. This makes it hard to justify whether the routing specificity metric is suitable for revealing the routing accessibility of representation-specific reasoning pathways.

[W4] The paper falls short of grounding the abstraction alignment method, raised in Section 4, within the relevant literature, since abstraction learning is not a novel idea and similar methods such as CoA [1], AoT [2] and AbstRaL [3] exist. A systematic comparison of the raised method to the related work would be necessary.

[W5] The findings of this paper are mainly limited to the satisfiability (SAT) reasoning domain and a single seed LLM (Qwen3-30B-A3B), making it questionable whether this work could be generalized to broader domains, such as the various math problems (formulated in FL and NL) in FormalMATH [4], and bigger or smaller LLMs.

[1] Gao et al., 2024. Efficient Tool Use with Chain-of-Abstraction Reasoning.
[2] Hong et al., 2024. Abstraction-of-Thought Makes Language Models Better Reasoners.
[3] Gao et al., 2025. AbstRaL: Augmenting LLMs' Reasoning by Reinforcing Abstract Thinking.
[4] Yu et al., 2025. FormalMATH: Benchmarking Formal Mathematical Reasoning of Large Language Models.

---

> ### Author Rebuttal · Authors · 2026-03-31
>
> We thank the reviewer for the thoughtful and constructive feedback. Below we address the main concerns.
>
> - NL–FL difference beyond surface form (W1 & Q1)
>
> We agree NL and FL differ beyond surface form. Our claim is narrower: this difference alone cannot explain the compartmentalization. The Knights-and-Knaves (K&K) benchmark provides the critical test — each constraint is a direct, unambiguous template statement with no narrative framing. If compartmentalization were driven by narrative complexity, it should attenuate on K&K. It does not.
>
> |Model|FL Acc.|NL Acc.|Jaccard|
> |-|-|-|-|
> |Qwen3 (Orig.)|85.43|93.29|0.0643|
> |GPT-oss (Orig.)|83.57|74.43|0.0884|
> |Qwen3 (RL)|89.29|93.71|0.1098|
> |GPT-oss (RL)|87.86|80.14|0.1164|
>
> Opposite FL/NL orderings with near-zero Jaccard rule out a difficulty account. FL-NL CKA decreases with complexity (62.08→51.66) while within-CKA stays stable (FL-FL: 89→87;
>   NL-NL: 83→82) — under fixed templates, this tracks logical structure, not surface form.
>
> |Model|FL-NL CKA (ppl=5)| (ppl=6)|(ppl=7)|(ppl=8)|Transfer Var.|Func-word Top-10 (FL/NL)|
> |-|-|-|-|-|-|-|
> |Qwen3 (Orig.)|62.08|54.79|53.00|51.66|15.68|4/5|
> |Qwen3 (RL)|62.12|54.78|52.95|51.55|11.48|4/4|
>
> Our central contribution is that this separation persists after alignment — accuracy and attribution improve, yet representational separation and weakly coupled dynamics remain, motivating the routing interpretation.
>
> - Reverse-direction learning dynamics (W2 & Q2)
>
> We trained exclusively on NL and tracked FL accuracy across 45 steps. The reverse trajectory shows the same hallmarks: large fluctuations, frequent sign changes (ΔFL/ΔNL ranges from −3.52 to +6.32 for Qwen3, −3.52 to +5.23 for GPT-oss), and no stable proportional relationship. Transfer variance (9.37/8.16) is lower than FL→NL (15.68/22.73), consistent with NL's lower baseline, but persistent sign changes confirm weakly coupled dynamics is bidirectional.
>
> - Clarification of routing specificity metric (W3 & Q3)
>
> At layer l, we replace NL hidden states with those from the isomorphic FL input and continue the forward pass. Δgold = log p_patch(gold) − log p_orig(gold); Δwrong = average change in log-prob of incorrect answers; routing specificity = Δgold − Δwrong. Matched: FL states from the isomorphic instance (same solution); Mismatched: from a different instance. Consistently higher matched specificity confirms content-specific reasoning information is routed, not generic FL properties.
>
> - Relation to prior abstraction methods (W4)
>
> We implemented an AoT-style baseline. AoT yields marginal gains, far below prompt-based and RL-based. Reasoning models bypass implicit abstraction; effective abstraction requires explicit structural output or verifiable reward.
>
> |Model|Original|AoT|Prompt-based|RL-based|
> |-|-|-|-|-|
> |Qwen3 (Bool avg.)|28.2|30.4|51.6|75.2|
>
> CoA uses underspecified placeholders for tool calls, not structural reconstruction. AbstRaL rewards answer correctness, functionally equivalent to training on NL with no constraint on abstraction itself; ours rewards isomorphism verification, directly supervising abstraction quality. The AoT results above confirm: without explicit structural enforcement, models bypass abstraction.
>
> - Generalizability across models and domains (W5)
>
> Table 1 already spans five LLMs (Qwen3-30B-A3B, GPT-oss-20B, DeepSeek-R1, Gemini-2.5-Pro, GPT-o3) with consistent FL→NL degradation. For DeepSeek-R1, lighter mechanistic diagnostics replicate the core pattern:
>
> |Mean FL-FL/NL-NL/FL-NL CKA|Func-word Top-10 (FL/NL)|
> |-|-|
> |0.709/0.768/0.059|3/4|
>
> For domain generalization beyond SAT/CSP, we evaluate on R2ATA (adversarial typographic perturbations on GSM8K, BBH, and MMLU) and CatAttack (adversarial triggers on NuminaMath):
>
> |Model|R2ATA Clean/Perturbed|CatAttack Clean/Perturbed|
> |-|-|-|
> |Qwen3 Orig→RL|90.47/86.87→91.18/87.50|96.50/94.50→96.50/96.00|
> |GPT-oss Orig→RL|79.18/70.81→82.41/76.11|63.00/61.00→70.60/70.00|
>
> Consistent gains indicate generalization beyond SAT/CSP. FormalMATH's FL–NL relation is autoformalization rather than logical isomorphism, but its finding that NL degrades formal proof performance is independently consistent with compartmentalization.
>
> - Joint FL+NL training (Q4)
>
> We trained jointly on FL and NL solving instances. Despite moderate NL gains, all compartmentalization signatures persist:
>
> |Diagnostic|Original|Joint-Trained|RL-Aligned|
> |-|-|-|-|
> |NL Perf. (Bool avg./CSP/Abel)|28.2/36.4/62.6|67.4/78.8/79.6|75.2/80.2/83.2|
> |FL-NL CKA|3.91|3.93|3.88|
> |Transfer Var.|15.68|14.72|11.48|
> |Func-word Top-10 (FL/NL)|4/5|4/5|4/4|
>
> Activation patching confirms matched > mismatched specificity peaking at layers 40–44 (gap 0.47–0.55). Joint training provides co-exposure but not representational fusion; abstraction alignment succeeds because it enforces explicit reconstruction of shared logical form — direct supervision that joint training lacks.

---

> > ### Author Rebuttal · Reviewer_8hAt · 2026-04-02
> >
> > Almost all my concerns have been addressed.
> >
> > For W4: The authors mention that their proposed abstraction alignment method can directly supervise abstraction quality by rewarding isomorphism verification. AbstRaL also rewards the quality of predicted abstraction, by measuring a symbolic edit distance to the gold abstraction. So the idea of doing abstraction alignment learning isn't totally novel. However, all the other concerns are well-resolved, and a major part of this paper also brings out new insights on the interpretability of LLM reasoning beyond accuracy improving.
> >
> > I will raise my score accordingly.

---

> > > ### Author Response · Authors · 2026-04-03
> > >
> > > We thank the reviewer for the correction regarding AbstRaL's reward design. We acknowledge that AbstRaL does supervise abstraction quality via a symbolic distance reward comparing generated abstractions to a gold reference, and our original characterization was inaccurate.
> > >
> > > To directly address this concern, we implemented AbstRaL on our task with both reward components: a ground-truth answer reward and a symbolic distance reward on FL. Results are as follows:
> > >
> > > | Model | Problem | Original | AbstRaL | AbstRaL (w/o sym. dist.) | Prompt-based | RL-based |
> > > |-|-|-|-|-|-|-|
> > > | Qwen3 | Bool | 28.2 | 32.6 | 40.4 | 51.6 | 75.2 |
> > > | Qwen3 | CSP | 36.4 | 60.4 | 76.0 | 66.2 | 80.2 |
> > > | Qwen3 | Abel | 62.6 | 66.0 | 82.2 | 77.8 | 83.2 |
> > >
> > > AbstRaL does improve over the original model, confirming that supervised abstraction alignment is a valid direction. However, removing the symbolic distance reward from AbstRaL consistently improves performance across all settings (Bool: 32.6→40.4, CSP: 60.4→76.0, Abel: 66.0→82.2), indicating that the symbolic distance reward introduces a conflicting training signal in our setting.
> > >
> > > This points to a fundamental difference in reward design assumptions. AbstRaL's symbolic distance reward implicitly assumes that the gold abstraction has a unique, canonical surface form — an assumption that holds in GSM-style problems where abstraction reduces to deterministic symbol substitution (e.g., replacing numbers with in0, in1 yields exactly one correct template). In our setting, however, a single NL problem admits multiple logically equivalent FL representations:
> > >
> > > ```
> > > gold FL:    Not(And(Not(A1), Not(A2)))
> > > generated:  Or(A1, A2) ← logically equivalent, but surface-distinct
> > > ```
> > >
> > > In such cases, symbolic distance conflates surface dissimilarity with semantic incorrectness, producing a spurious training signal. Our isomorphism verification reward addresses this by directly verifying semantic equivalence, accepting any logically equivalent FL form regardless of surface representation. This makes it applicable to a strictly broader class of abstraction tasks: when abstractions do have a unique surface form, isomorphism verification reduces to exact match; when they do not, it remains correct while symbolic distance breaks down.
> > >
> > > We therefore argue that isomorphism verification represents a more general reward design principle for abstraction alignment, of which surface-level distance is a special case. The ablation above provides direct empirical support for this claim.

---

### Decision · Program_Chairs · 2026-04-30

**Decision:**

Accept (regular)

**Comment:**

The paper addresses formulation sensitivity in LLM reasoning using a controlled FL–NL isomorphic setup. Reviewers found the problem interesting and the paper technically solid, emphasizing the clean design, the framing of reasoning compartmentalization, and the mechanistic evidence from CKA and activation patching. The proposed abstraction alignment method was viewed as effective, with evidence suggesting gains come from improved routing rather than representational fusion.

Main concerns involved external validity, possible alternative explanations related to natural-language complexity and data construction, some presentation issues, and overlap with prior abstraction-based methods. The rebuttal addressed these with new evidence, including K&K results, reverse-direction transfer, stronger baselines and ablations, sensitivity analyses, clarification of the routing metric, and broader out-of-domain evaluations. Most reviewers indicated that their concerns were resolved and maintained or increased their scores. One reviewer remained somewhat unconvinced about the root cause and scope of the mitigation, but the overall consensus was positive. One remaining issue is that the submission does not currently provide a code repository link to ensure full reproducibility check.  Acceptance is recommended, conditional on the authors providing a code repository link in the final camera-ready version as a requirement for full reproducibility of the results